# A Stochastic Trust Region Method for Non-convex Minimization

## Abstract

We target the problem of finding a local minimum in non-convex finite-sum minimization. Towards this goal, we first prove that the trust region method with inexact gradient and Hessian estimation can achieve a convergence rate of order $\mathcal{O}(1/k^{2/3})$ as long as those differential estimations are sufficiently accurate. Combining such result with a novel Hessian estimator, we propose a sample-efficient stochastic trust region (STR) algorithm which finds an $(\epsilon, \sqrt{\epsilon})$-approximate local minimum within $\tilde{\mathcal{O}}(\sqrt{n}/\epsilon^{1.5})$ stochastic Hessian oracle queries. This improves the state-of-the-art result by a factor of $\mathcal{O}(n^{1/6})$. Finally, we also develop Hessian-free STR algorithms which achieve the lowest runtime complexity. Experiments verify theoretical conclusions and the efficiency of the proposed algorithms.

## 1 Introduction

We consider the following finite-sum non-convex minimization problem

$$\min_{\mathbf{x} \in \mathbb{R}^d} F(\mathbf{x}) = \frac{1}{n} \sum_{i=1}^{n} f_i(\mathbf{x}), \tag{1}$$

where each (non-convex) component function $f_i : \mathbb{R}^d \rightarrow \mathbb{R}$ is assumed to have $L_1$-Lipschitz continuous gradient and $L_2$-Lipschitz continuous Hessian. Since first-order stationary points could be saddle points with inferior generalization performance (Dauphin et al., 2014), in this work we are particularly interested in computing $(\epsilon, \sqrt{\epsilon})$-approximate second-order stationary points, $\epsilon$-SOSP:

$$\|\nabla F(\mathbf{x}_\epsilon)\| \leq \epsilon \quad \text{and} \quad \nabla^2 F(\mathbf{x}_\epsilon) \succcurlyeq -\sqrt{L_2 \epsilon} \mathbf{I}. \tag{2}$$

To find a local minimum of problem (1), the cubic regularization approach (Nesterov & Polyak, 2006) and the trust region algorithm (Conn et al., 2000; Curtis et al., 2017) are two classical methods. Specifically, cubic regularization forms a cubic surrogate function for the objective $F(\mathbf{x})$ by adding a third-order regularization term to the second-order Taylor expansion, and minimizes it iteratively. Such a method is proved to achieve an $\mathcal{O}(1/k^{2/3})$ global convergence rate and thus needs $\mathcal{O}(n/\epsilon^{1.5})$ stochastic first- and second-order oracle queries, namely the evaluation number of stochastic gradient and Hessian, to achieve a point that satisfies (2). On the other hand, trust region algorithms estimate the objective with its second-order Taylor expansion but minimize it only within a local region. Recently, Curtis et al. (2017) proposes a trust region variant to achieve the same convergence rate as the cubic regularization approach. But both methods require computing full gradients and Hessians of $F(\mathbf{x})$ and thus suffer from high computational cost in large-scale problems.

To avoid costly exact differential evaluations, many works explore the finite-sum structure of problem (1) and develop stochastic cubic regularization approaches. Both Kohler & Lucchi (2017b) and Xu et al. (2017) propose to directly subsample the gradient and Hessian in the cubic surrogate function, and achieve $\tilde{\mathcal{O}}(1/\epsilon^{3.5})$ and $\tilde{\mathcal{O}}(1/\epsilon^{2.5})$ stochastic first- and second-order oracle complexities respectively. By plugging a stochastic variance reduced estimator (Johnson & Zhang, 2013) and the Hessian tracking technique (Gower et al., 2018) into the gradient and Hessian estimation, the approach in (Zhou et al., 2018a) improves both the stochastic first- and second-order oracle complexities to $\tilde{\mathcal{O}}(n^{0.8}/\epsilon^{1.5})$. Recently, Zhang et al. (2018) and Zhou et al. (2018b) develop more efficient stochastic cubic regularization variants, which further reduce the stochastic second-order oracle complexity to $\tilde{\mathcal{O}}(n^{2/3}/\epsilon^{1.5})$ at the cost of increasing the stochastic first-order oracle complexity to $\tilde{\mathcal{O}}(n^{2/3}/\epsilon^{2.5})$.

Table 1: Stochastic first- and second-order oracle complexities, SFO and SSO for short respectively, of the proposed STR approaches and other state-of-the-art methods. When SSO is prioritized, our $STR_1$ has strictly better complexity than both SCR and Lite-SVRC. When SFO and SSO are treated equally, $STR_2$ improves the existing result in SVRC.

| Algorithm | SFO | SSO |
|---|---|---|
| TR (Conn et al., 2000; Curtis et al., 2017) | $\mathcal{O}(\frac{n}{\epsilon^{1.5}})$ | $\mathcal{O}(\frac{n}{\epsilon^{1.5}})$ |
| CR (Nesterov & Polyak, 2006) | $\mathcal{O}(\frac{n}{\epsilon^{1.5}})$ | $\mathcal{O}(\frac{n}{\epsilon^{1.5}})$ |
| SCR (Kohler & Lucchi, 2017a) | $\tilde{\mathcal{O}}(\frac{1}{\epsilon^{3.5}})$ | $\tilde{\mathcal{O}}(\frac{1}{\epsilon^{2.5}})$ |
| SVRC (Zhou et al., 2018c) | $\tilde{\mathcal{O}}(\frac{n^{4/5}}{\epsilon^{1.5}})$ | $\tilde{\mathcal{O}}(\frac{n^{4/5}}{\epsilon^{1.5}})$ |
| Lite-SVRC (Zhou et al., 2018b) | $\tilde{\mathcal{O}}(\frac{n^{2/3}}{\epsilon^{2.5}})$ | $\tilde{\mathcal{O}}(\frac{n^{2/3}}{\epsilon^{1.5}})$ |
| $STR_1$ (this paper) | $\tilde{\mathcal{O}}(\min\{\frac{n}{\epsilon^{1.5}}, \frac{\sqrt{n}}{\epsilon^2}\})$ | $\tilde{\mathcal{O}}(\min\{\frac{1}{\epsilon^2}, \frac{\sqrt{n}}{\epsilon^{1.5}}\})$ |
| $STR_2$ (this paper) | $\tilde{\mathcal{O}}(\frac{n^{3/4}}{\epsilon^{1.5}})$ | $\tilde{\mathcal{O}}(\frac{n^{3/4}}{\epsilon^{1.5}})$ |

**Contributions:** In this paper we propose and exploit a formulation in which we make explicit control of the step size in the trust region method. This idea is leveraged to develop two efficient stochastic trust region (STR) approaches. We tailor our methods to achieve state-of-the-art oracle complexities under the following two measurements: (i) the stochastic second-order oracle complexity is prioritized; (ii) the stochastic first- and second-order oracle complexities are treated equally. Specifically, in Setting (i), our method $STR_1$ employs a newly proposed estimator to approximate the Hessian and adopts the estimator in (Fang et al., 2018) for gradient approximation. Our novel Hessian estimator maintains an accurate second-order differential approximation with lower amortized oracle complexity. In this way, $STR_1$ achieves $\tilde{\mathcal{O}}(\min\{1/\epsilon^2, \sqrt{n}/\epsilon^{1.5}\})$ stochastic second-order oracle complexity. This is lower than existing results for solving problem (1). In Setting (ii), our method $STR_2$ substitutes the gradient estimator in $STR_1$ with one that integrates stochastic gradient and Hessian together to maintain an accurate gradient approximation. As a result, $STR_2$ achieves convergence in $\tilde{\mathcal{O}}(n^{3/4}/\epsilon^{1.5})$ overall stochastic first- and second-order oracle queries. Finally, based on STR, we further develop Hessian-free STR algorithms, namely $STR_{free}$ and $STR_{free}+$, which outperform existing Hessian-free algorithms theoretically.

## 1.1 RELATED WORK

Computing local minimum to a non-convex optimization problem is gaining considerable amount of attentions in recent years. Both cubic regularization (CR) approaches (Nesterov & Polyak, 2006) and trust region (TR) algorithms (Conn et al., 2000; Curtis et al., 2017) can escape saddle points and find a local minimum by iterating the variable along the direction related to the eigenvector of the Hessian with the most negative eigenvalue. As the CR heavily depends on the regularization parameter for the cubic term, Cartis et al. (2011) propose an adaptive cubic regularization (ARC) approach to boost the efficiency by adaptively tuning the regularization parameter according to the current objective decrease. Noting the high cost of full gradient and Hessian computation in ARC, sub-sampled cubic regularization (SCR) (Kohler & Lucchi, 2017a) is developed for sampling partial data points to estimate the full gradient and Hessian. Recently, by exploring the finite-sum structure of the target problem, many works incorporate variance reduction technique (Johnson & Zhang, 2013) into CR and propose stochastic variance-reduced methods. For example, Zhou et al. (2018c) propose stochastic variance-reduced cubic (SVRC) in which they integrate the stochastic variance-reduced gradient estimator (Johnson & Zhang, 2013) and the Hessian tracking technique (Gower et al., 2018) with CR. Such a method is proved to be at least $\mathcal{O}(n^{1/5})$ faster than CR and TR. Then Zhou et al. (2018b) use adaptive gradient batch size and constant Hessian batch size, and develop Lite-SVRC to further reduce the stochastic second-order oracle $\tilde{\mathcal{O}}(n^{4/5}/\epsilon^{1.5})$ of SVRC to $\tilde{\mathcal{O}}(n^{2/3}/\epsilon^{1.5})$ at the cost of higher gradient computation cost. Similarly, except turning the gradient batch size, Zhang et al. (2018) further adaptively sample a certain number of data points to estimate the Hessian and prove the proposed method to have the same stochastic second-order oracle complexity as Lite-SVRC.

## 2 PRELIMINARY

**Notation.** We use $\|\mathbf{v}\|$ to denote the Euclidean norm of vector $\mathbf{v}$ and use $\|\mathbf{A}\|$ to denote the spectral norm of matrix $\mathbf{A}$. Let $\mathcal{S}$ be the set of component indices. We define the minibatch average of component functions by $f(\mathbf{x}; \mathcal{S}) \overset{\text{def}}{=} \frac{1}{|\mathcal{S}|} \sum_{i \in \mathcal{S}} f_i(\mathbf{x})$. Then we specify the assumptions that are necessary to the analysis of our methods.

---

**MetaAlgorithm 1** Inexact Trust Region Method

---

**Input:** initial point $\mathbf{x}^0$, step size $r$, number of iterations $K$, construction of differential estimators $\mathbf{g}^k$ and $\mathbf{H}^k$
1: **for** $k = 0$ **to** $K - 1$ **do**
2:      Compute $\mathbf{h}^k$ and $\lambda^k$ by solving (8);
3:      $\mathbf{x}^{k+1} := \mathbf{x}^k + \mathbf{h}^k$;
4:      **if** $\lambda^k \leq 3\sqrt{\epsilon/L_2}$ **then**
5:         Output $\mathbf{x}_\epsilon = \mathbf{x}^{k+1}$;
6:      **end if**
7: **end for**

---

**Assumption 2.1.** *F is bounded from below and its global optimal is achieved at $\mathbf{x}^*$. We denote $\Delta = F(\mathbf{x}^0) - F(\mathbf{x}^*)$.*

**Assumption 2.2.** *Each $f_i : \mathbb{R}^d \to \mathbb{R}$ has $L_1$-Lipschitz continuous gradient: for any $\mathbf{x}, \mathbf{y} \in \mathbb{R}^d$*

$$\|\nabla f_i(\mathbf{x}) - \nabla f_i(\mathbf{y})\| \leq L_1 \|\mathbf{x} - \mathbf{y}\|. \tag{3}$$

**Assumption 2.3.** *Each $f_i : \mathbb{R}^d \to \mathbb{R}$ has $L_2$-Lipschitz continuous Hessian: for any $\mathbf{x}, \mathbf{y} \in \mathbb{R}^d$*

$$\|\nabla^2 f_i(\mathbf{x}) - \nabla^2 f_i(\mathbf{y})\| \leq L_2 \|\mathbf{x} - \mathbf{y}\|. \tag{4}$$

## 2.1 TRUST REGION METHOD

Here we briefly introduce the trust region method (Conn et al., 2000). In each step, it first solves the Quadratic Constraint Quadratic Program (QCQP) defined as

$$\mathbf{h}^k := \underset{\mathbf{h} \in \mathbb{R}^d, \|\mathbf{h}\| \leq r}{\operatorname{argmin}} \langle \nabla F(\mathbf{x}^k), \mathbf{h} \rangle + \frac{1}{2} \langle \nabla^2 F(\mathbf{x}^k)\mathbf{h}, \mathbf{h} \rangle, \tag{5}$$

where $r$ is the trust-region radius. Then it updates the new variable as

$$\mathbf{x}^{k+1} := \mathbf{x}^k + \mathbf{h}^k. \tag{6}$$

Since $\nabla^2 F(\mathbf{x}^k)$ is indefinite, the trust-region subproblem (5) is non-convex. But its global optimizer can be characterized by the following lemma (Corollary 7.2.2 in (Conn et al., 2000)).

**Lemma 2.1.** *Any global minimizer of problem (5) satisfies the equation*

$$\left( \nabla^2 F(\mathbf{x}^k) + \lambda \mathbf{I} \right) \mathbf{h}^k = -\nabla F(\mathbf{x}^k), \tag{7}$$

*where the dual variable $\lambda \geq 0$ should satisfy $\nabla^2 F(\mathbf{x}^k) + \lambda \mathbf{I} \succcurlyeq 0$ and $\lambda(\|\mathbf{h}^k\| - r) = 0$.*

In particular, the standard QCQP solver returns both the minimizer $\mathbf{h}^k$ as well as the corresponding dual variable $\lambda$ of subproblem (5). In the following section, we first prove that the deterministic trust-region update (5) and (6) converges at the rate of $\mathcal{O}(1/k^{2/3})$, much sharper than existing provable convergence rate $\mathcal{O}(1/\sqrt{k})$ (Conn et al., 2000), and then develop a more efficient stochastic trust-region approach.

## 3 METHODOLOGY

Here we first introduce a general inexact trust region method which is summarized in MetaAlgorithm 1. It accepts inexact gradient estimation $\mathbf{g}^k$ and Hessian estimation $\mathbf{H}^k$ as input to the QCQP subproblem

$$\mathbf{h}^k := \underset{\mathbf{h} \in \mathbb{R}^d, \|\mathbf{h}\| \leq r}{\operatorname{argmin}} \langle \mathbf{g}^k, \mathbf{h} \rangle + \frac{1}{2} \langle \mathbf{H}^k \mathbf{h}, \mathbf{h} \rangle. \tag{8}$$

Similar to (5), Lemma 2.1 characterizes the global optimum to problem (8) which can be efficiently solved by Lanczos method (Gould et al., 1999). Assume the dual variable of the minimizer $\mathbf{h}^k$ is $\lambda^k$.

We prove that such inexact trust region method achieves the optimal $\mathcal{O}(1/k^{2/3})$ convergence rate when the estimation $\mathbf{g}^k$ and $\mathbf{H}^k$ at each iteration are sufficiently close to their full (exact) counterparts $\nabla F(\mathbf{x}^k)$ and $\nabla^2 F(\mathbf{x}^k)$ respectively:

$$\|\mathbf{g}^k - \nabla F(\mathbf{x}^k)\| \leq \frac{\epsilon}{6}, \quad \|\mathbf{H}^k - \nabla^2 F(\mathbf{x}^k)\| \leq \frac{\sqrt{\epsilon L_2}}{3}. \tag{9}$$

---

**Algorithm 2** $STR_1$

---

**Input:** initial point $\mathbf{x}^0$, step size $r$, number of iterations $K$
1: **for** $k = 1$ **to** $K$ **do**
2:       Construct gradient estimator $\mathbf{g}^k$ by Estimator 4;
3:       Construct Hessian estimator $\mathbf{H}^k$ by Estimator 3;
4:       Compute $\mathbf{h}^k$ and $\lambda^k$ by solving (8);
5:       $\mathbf{x}^{k+1} := \mathbf{x}^k + \mathbf{h}^k$;
6:       **if** $\lambda^k \leq 3\sqrt{\epsilon/L_2}$ **then**
7:           Output $\mathbf{x}_\epsilon = \mathbf{x}^{k+1}$;
8:       **end if**
9: **end for**

---

Such result allows us to derive stochastic trust-region variants with novel differential estimators that are tailored to ensure the optimal convergence rate. We state our formal results in Theorem 3.1, whose proof is deferred to Appendix B.1 due to the space limit.

**Theorem 3.1** (Main Result). *Consider problem (1) under Assumption 2.1-2.3. If the differential estimators $\mathbf{g}^k$ and $\mathbf{H}^k$ satisfy Eqn. (9) for all $k$, MetaAlgorithm 1 finds an $\mathcal{O}(\epsilon)$-SOSP in less than $K = \mathcal{O}(\sqrt{L_2}\Delta/\epsilon^{1.5})$ iterations by setting the trust-region radius as $r = \sqrt{\epsilon/L_2}$.*

**Remark 3.1.** *We emphasize that MetaAlgorithm 1 degenerates to the exact trust region method by taking $\mathbf{g}^k = \nabla F(\mathbf{x}^k)$ and $\mathbf{H}^k = \nabla^2 F(\mathbf{x}^k)$. Such result is of its own interest because this is the first proof to show that the vanilla trust region method has the optimal $\mathcal{O}(1/k^{2/3})$ convergence rate. Similar rate is achieved by Curtis et al. (2017) but with a much more complicated trust region variant.*

Theorem 3.1 shows the explicit step size control of the trust region method: Since the dual variable satisfies $\lambda^k > 3\epsilon^{0.5}/\sqrt{L_2} > 0$ for all but the last iteration, we always find the solution to the trust-region subproblem (8) in the boundary, i.e. $\|\mathbf{h}^k\| = r$, according to the complementary condition (15) in Appendix B.1. Such exact step size control property is missing in the cubic-regularization method where the step size is implicitly decided by the cubic regularization parameter.

More importantly, we emphasize that such explicit step size control is crucial to the sample efficiency of our variance reduced differential estimators. The essence of variance reduction is to exploit the correlations between the differentials in consecutive iterations. Intuitively, when two neighboring iterates are close, so are their differentials due to the Lipschitz continuity, and hence a smaller number of samples suffice to maintain the accuracy of the estimators. On the other hand, smaller step size reduces the per-iteration objective decrease which harms the convergence rate of the algorithm (see proof of Theorem 3.1). Therefore, the explicit step size control in trust region method allows us to well trade-off the per-iteration sample complexity and convergence rate, from which we can derive stochastic trust region approaches with the state-of-the-art sample efficiency. In contrast, existing trust region methods change the step size at every iteration according to progress made, which requires loss evaluations that can be as expensive as gradient computations (e.g. the non-convex linear model in Section 7) and is thus prohibitive for large-scale problems.

## 4    STOCHASTIC TRUST REGION METHOD: TYPE I

Having the inexact trust region method as prototype, we now present our first sample-efficient stochastic trust region method, namely $STR_1$, in Algorithm 2 which emphasizes cheaper stochastic second-order oracle complexity. As Theorem 3.1 already guarantees the optimal convergence rate of MetaAlgorithm 1 when the gradient estimator $\mathbf{g}^k$ and the Hessian estimator $\mathbf{H}^k$ meet the requirement (9), here we focus on constructing such novel differential estimators. Specifically, we first present our Hessian estimator in Estimator 3 and our first gradient estimator in Estimator 4, both of which exploit the trust region radius $r = \sqrt{\epsilon/L_2}$ to reduce their variances.

### 4.1    HESSIAN ESTIMATOR

Our epoch-wise Hessian estimator $\mathbf{H}^k$ is given in Estimator 3, where $p_2$ controls the epoch length and $s_2$ (and optionally $s_2'$) controls the minibatch size. At the beginning of each epoch, Estimator 3 has two options, designed for different target accuracy: Option I is preferable for the high accuracy

---

**Estimator 3** Hessian Estimator

**Input:** Epoch length $p_2$, sample size $s_2$, $s_2'$ (optional)
1: **if** $\mod(k, p_2) = 0$ **then**
2:     Option I:     ⋄ high accuracy case (small $\epsilon$)
3:     $\mathbf{H}^k := \nabla^2 F(\mathbf{x}^k)$;
4:     Option II:     ⋄ low accuracy case (moderate $\epsilon$)
5:     Draw $s_2'$ samples indexed by $\mathcal{H}'$ and let $\mathbf{H}^k := \nabla^2 f(\mathbf{x}^k; \mathcal{H}')$;
6: **else**
7:     Draw $s_2$ samples indexed by $\mathcal{H}$ and let $\mathbf{H}^k := \nabla^2 f(\mathbf{x}^k; \mathcal{H}) - \nabla^2 f(\mathbf{x}^{k-1}; \mathcal{H}) + \mathbf{H}^{k-1}$;
8: **end if**

---

**Estimator 4** Gradient Estimator: Case (1)

1: **if** $\mod(k, p_1) = 0$ **then**
2:     $\mathbf{g}^k := \nabla F(\mathbf{x}^k)$;
3: **else**
4:     Draw $s_1$ samples indexed by $\mathcal{G}$ and $\mathbf{g}^k = \nabla f(\mathbf{x}^k; \mathcal{G}) - \nabla f(\mathbf{x}^{k-1}; \mathcal{G}) + \mathbf{g}^{k-1}$;
5: **end if**

---

case ($\epsilon < \mathcal{O}(1/n)$) where we compute the full Hessian to avoid approximation error, and Option II is designed for the moderate accuracy case ($\epsilon > \mathcal{O}(1/n)$) where we only need an approximate Hessian estimator. Then, $p_2$ iterations follow with $\mathbf{H}^k$ defined in a recurrent manner. These recurrent estimators exist for the first-order case (Nguyen et al., 2017; Fang et al., 2018), but their bound only holds under the vector $\ell_2$ norm. Here we generalize them into Hessian estimation with matrix spectral norm bound.

The following lemma analyzes the amortized stochastic second-order oracle (Hessian) complexity for Algorithm 3 to meet the requirement in Theorem 3.1. As we need an error bound under the spectral norm, we will appeal to the matrix Azuma's inequality (Tropp, 2012). The proof is deferred to Appendix B.2.

**Lemma 4.1.** *Assume Algorithm 2 takes the trust region radius $r = \sqrt{\epsilon/L_2}$ as in Theorem 3.1. For any $k \geq 0$, Estimator 3 produces estimator $\mathbf{H}^k$ for the second order differential $\nabla^2 F(\mathbf{x}^k)$ such that $\|\mathbf{H}^k - \nabla^2 F(\mathbf{x}^k)\| \leq \sqrt{\epsilon L_2}/3$ with probability at least $1 - \delta/K_0$ if we set (1) $p_2 = \sqrt{n}$ and $s_2 = 32\sqrt{n}\log(dK_0/\delta)$ in option I, or (2) $p_2 = L_1/(2\sqrt{\epsilon L_2})$, $s_2' = 16L_1^2/(\epsilon L_2)\log(dK_0/\delta)$, and $s_2 = 32L_1/(\sqrt{\epsilon L_2})\log(dK_0/\delta)$ in option II. Here $K_0$ is a constant to be determined later. Consequently the amortized per-iteration stochastic second-order oracle complexity to construct $\mathbf{H}^k$ is no more than $2s_2 = \min\{64\sqrt{n}\log\frac{dK_0}{\delta}, \frac{64L_1}{\sqrt{\epsilon L_2}}\log\frac{dK_0}{\delta}\}$.*

### 4.2   Gradient Estimator: Case (1)

When the stochastic second-order oracle complexity is prioritized, we directly employ the SPIDER gradient estimator to construct $\mathbf{g}^k$ (Fang et al., 2018). Similar to the construction for $\mathbf{H}^k$, the estimator $\mathbf{g}^k$ is also constructed in an epoch-wise manner as presented in Estimator 4, where $p_1$ controls the epoch length and $s_1$ controls the minibatch size.

We now analyze the stochastic first-order oracle complexity to meet the requirement in Theorem 3.1.

**Lemma 4.2.** *Assume Algorithm 2 takes the trust region radius $r = \sqrt{\epsilon/L_2}$. Estimator 4 produces estimator $\mathbf{g}^k$ of the first order differential $\nabla F(\mathbf{x}^k)$ such that $\|\mathbf{g}^k - \nabla F(\mathbf{x}^k)\| \leq \epsilon/6$ with probability at least $1 - \delta/K_0$ for any $k \geq 0$, if we set $p_1 = \max\{1, \sqrt{n\epsilon L_2/(cL_1^2\log\frac{K_0}{\delta})}\}$ and $s_1 = \min\{n, \sqrt{cnL_1^2\log(K_0/\delta)/(\epsilon L_2)}\}$, where the constant $c = 1152$ and $K_0$ is a constant to be determined later. Consequently, the amortized per-iteration stochastic first-order oracle complexity to construct $\mathbf{g}^k$ is $\min\{n, \sqrt{4cnL_1^2\log(K_0/\delta)/(\epsilon L_2)}\}$.*

The proof of Lemma 4.2 is similar to the one of Lemma 4.1 and is deferred to Appendix B.3. These two lemmas only guarantee that the differential estimators satisfy the requirement (9) in a single iteration and can be extended to hold for all $k$ by using the union bound with $K_0 = 2K$, where $K$ denotes the number of iterations. Combining such lifted result with Theorem 3.1, we can establish the computational complexity bound as follows.

---

**Algorithm 5** STR$_2$

---

**Input:** initial point $\mathbf{x}^0$, step size $r$, number of iterations $K$
1: **for** $k = 1$ **to** $K$ **do**
2:     Construct gradient estimator $\mathbf{g}^k$ by Estimator 6;
3:     Construct Hessian estimator $\mathbf{H}^k$ by Estimator 3;
4:     Compute $\mathbf{h}^k$ and $\lambda^k$ by solving (8);
5:     $\mathbf{x}^{k+1} := \mathbf{x}^k + \mathbf{h}^k$;
6:     **if** $\lambda^k \leq 3\sqrt{\epsilon/L_2}$ **then**
7:         Output $\mathbf{x}_\epsilon = \mathbf{x}^{k+1}$;
8:     **end if**
9: **end for**

---

**Estimator 6** Gradient Estimator: Case (2)

---

1: **if** $\mathrm{mod}(k, p_1) = 0$ **then**
2:     Let $\tilde{\mathbf{x}} := \mathbf{x}^k$, $\mathbf{g}^k := \nabla F(\tilde{\mathbf{x}})$
3: **else**
4:     Draw $s_1$ samples indexed by $\mathcal{G}$;
5:     $\mathbf{g}^k = \nabla f(\mathbf{x}^k; \mathcal{G}) - \nabla f(\mathbf{x}^{k-1}; \mathcal{G}) + \mathbf{g}^{k-1} + [\nabla^2 F(\tilde{\mathbf{x}}) - \nabla^2 f(\tilde{\mathbf{x}}; \mathcal{G})](\mathbf{x}^k - \mathbf{x}^{k-1})$;
6: **end if**

---

**Corollary 4.1** (Computational Complexity of STR$_1$)**.** *Assume Algorithm 2 will use Estimator 4 to construct the first-order differential estimator $\mathbf{g}^k$ and use Estimator 3 to construct the second-order differential estimator $\mathbf{H}^k$. To find a $12\epsilon$-SOSP with probability at least $1 - \delta$, the overall stochastic first-order oracle complexity is $\mathcal{O}(\min\{\frac{n\sqrt{L_2}\Delta}{\epsilon^{1.5}}, \frac{\sqrt{n}L_1}{\epsilon^2}\log(\frac{L_2\Delta}{\delta\epsilon^{1.5}})\})$ and the overall stochastic second-order oracle complexity is $\mathcal{O}(\min\{\frac{\sqrt{nL_2}\Delta}{\epsilon^{1.5}}, \frac{L_1\Delta}{\epsilon^2}\}\log(\frac{d\sqrt{L_2}\Delta}{\delta\epsilon^{1.5}}))$.*

From Corollary 4.1 we see that $\tilde{\mathcal{O}}(\min\{\sqrt{n}/\epsilon^{1.5}, 1/\epsilon^2\})$ stochastic second-order oracle queries are sufficient for STR$_1$ to find an $\epsilon$-SOSP which is significantly better than both the subsampled cubic regularization method $\tilde{\mathcal{O}}(1/\epsilon^{2.5})$ (Kohler & Lucchi, 2017a) and the variance reduction based ones $\tilde{\mathcal{O}}(n^{2/3}/\epsilon^{1.5})$ (Zhou et al., 2018b; Zhang et al., 2018). Recently, Zhou & Gu (2019) developed a stochastic recursive variance-reduced cubic regularization (SRVRC) method which finds an $(\epsilon, \sqrt{\epsilon})$-approximate local minimum with $\tilde{\mathcal{O}}(n/\epsilon^{1.5}, 1/\epsilon^3)$ SFO and $\tilde{\mathcal{O}}(\sqrt{n}/\epsilon^{1.5}, 1/\epsilon^2)$ SSO. But the result of SRVRC needs to assume stochastic gradient to be bounded, i.e., $\|\nabla f_i(\mathbf{x}) - \nabla F(\mathbf{x})\| \leq \sigma$. With this extra assumption, STR1 enjoys $\tilde{\mathcal{O}}(n/\epsilon^{1.5}, n/\epsilon^2, 1/\epsilon^3)$ SFO and $\tilde{\mathcal{O}}(\sqrt{n}/\epsilon^{1.5}, 1/\epsilon^2)$ SSO. Thus, if $1/\epsilon \leq n \leq 1/\epsilon^2$, STR1 outperforms SRVRC; otherwise they have the same complexities.

## 5    STOCHASTIC TRUST REGION METHOD: TYPE II

In the above section, we focus on the setting where the stochastic second-order oracle complexity is prioritized over the stochastic first-order oracle complexity. In this setting, STR$_1$ achieves the state-of-the-art efficiency. In this section, we consider a different complexity measure where first-order and second-order oracle complexities are treated equally and our goal is to minimize the maximum of them. We note that, currently the best result is $\tilde{\mathcal{O}}(n^{4/5}/\epsilon^{1.5})$ of the SVRC method (Zhou et al., 2018c).

Since the Hessian estimator $\mathbf{H}^k$ of STR$_1$ already delivers the superior $\tilde{\mathcal{O}}(\sqrt{n}/\epsilon^{1.5})$ stochastic Hessian complexity, in STR$_2$ (see Algorithm 5), we retain Estimator 3 for second-order differential estimation and use Estimator 6 to further reduce the stochastic gradient complexity.

### 5.1    GRADIENT ESTIMATOR: CASE (2)

When stochastic gradient and Hessian complexities are equally important, we use Hessian to improve the gradient estimation. Denote $\mathbf{x}(a) = a\mathbf{x}^t + (1 - a)\tilde{\mathbf{x}}$. From Assumption 2.3, we have

$$\|\nabla f_i(\mathbf{x}^t) - \nabla f_i(\tilde{\mathbf{x}}) - \nabla^2 f_i(\tilde{\mathbf{x}})(\mathbf{x}^t - \tilde{\mathbf{x}})\| = \left\|\int_0^1 [\nabla^2 f_i(\mathbf{x}(a)) - \nabla^2 f_i(\tilde{\mathbf{x}})](\mathbf{x}^t - \tilde{\mathbf{x}})\mathrm{d}a\right\| \leq \frac{L_2}{2}\|\mathbf{x}^t - \tilde{\mathbf{x}}\|^2.$$

Such property can be used to improve Lemma 4.2 of Estimator 4. Specifically, define the correction

$$\mathbf{c}^k = [\nabla^2 F(\tilde{\mathbf{x}}) - \nabla^2 f(\tilde{\mathbf{x}}; \mathcal{G})](\mathbf{x}^k - \mathbf{x}^{k-1}),$$

where $\tilde{\mathbf{x}}$ is some reference point updated in an epoch-wise manner. Estimator 6 adds $\mathbf{c}^k$ to the estimator in Estimator 4. Note that in Estimator 6, the first- and second-order oracle complexities are the same. We now analyze the first-order (and second-order) oracle complexity to meet requirement (9).

**Lemma 5.1.** *Assume Algorithm 5 takes the trust region radius $r = \sqrt{\epsilon/L_2}$ as in Theorem 3.1. For any $k \geq 0$, Estimator 6 produces estimator $\mathbf{g}^k$ for the first order differential $\nabla F(\mathbf{x}^k)$ such that $\|\mathbf{g}^k - \nabla F(\mathbf{x}^k)\| \leq \epsilon/6$ with probability at least $1 - \delta/K_0$, if we set $p_1 = n^{0.25}$ and $s_1 = n^{0.75} c \log(K_0/\delta)$, where $c = 1152$ and $K_0$ is a constant to be determined. Consequently, the amortized per-iteration stochastic first-order oracle complexity to construct $\mathbf{g}^k$ is $2s_1 = 2n^{0.75} c \log(K_0/\delta)$.*

The proof of Lemma 5.1 is similar to the one of Lemma 4.1 and is deferred to Appendix B.4. Similar to the previous section, Lemma 5.1 only guarantees that the gradient estimator satisfies the requirement (9) in a single iteration. Such result can be extended to hold for all $k$ by using the union bound with $K_0 = 2K$, which together with Theorem 3.1 gives the following corollary.

**Corollary 5.1** (Computational Complexity of STR₂). *Algorithm 5 finds a $12\epsilon$-SOSP with probability at least $1 - \delta$, within $\mathcal{O}(\frac{n^{0.75}\sqrt{L_2}\Delta}{\epsilon^{1.5}} \log(\frac{\sqrt{L_2}\Delta}{\delta\epsilon^{1.5}}))$ overall stochastic first-order oracle queries and $\mathcal{O}(\frac{n^{0.75}\sqrt{L_2}\Delta}{\epsilon^{1.5}} \log(\frac{d\sqrt{L_2}\Delta}{\delta\epsilon^{1.5}}))$ overall stochastic second-order oracle queries.*

Corollary 5.1 shows that to find an $\epsilon$-SOSP, both SFO and SSO of STR₂ are $\tilde{\mathcal{O}}(n^{3/4}/\epsilon^{1.5})$ which surpasses the best existing one $\tilde{\mathcal{O}}(n^{4/5}/\epsilon^{1.5})$ in (Zhou et al., 2018c).

# 6 PRACTICAL STOCHASTIC TRUST REGION VARIANTS

## 6.1 HANDLING INEXACT QCQP SOLUTIONS

One drawback of MetaAlgorithm 1 is that it requires the exact solution to the QCQP subproblem (8) and uses the dual variable as stopping criterion. We address this problem by developing a practical variant, MetaAlgorithm 7, which admits inexact QCQP solutions without access to the dual variable. This algorithm repeatedly invokes a procedure called INEXACTTR_WEAK, which, as we shall see, outputs an $\mathcal{O}(\epsilon)$-SOSP with a constant probability of 2/3 in $\mathcal{O}(1/\epsilon^{1.5})$ iterations. By repeatedly invoking INEXACTTR_WEAK for $\Theta(\log(1/\delta))$ times, MetaAlgorithm 7 boosts the probability to $(1 - \delta)$ for any desired $\delta$. This repeating technique has been studied by, e.g., (Allen-Zhu & Li, 2018; Allen-Zhu, 2018b). To test whether the $t$-th run outputs an $\mathcal{O}(\epsilon)$-SOSP, we need to compute $\|\nabla F(\mathbf{x}^t)\|$ and the smallest eigenvalue of $\nabla^2 F(\mathbf{x}^t)$. The latter one can be approximated by solving the QCQP

$$\mathbf{v}^t := \underset{\|\mathbf{v}\| \leq 1}{\operatorname{argmin}} \psi_t(\mathbf{v}) = \langle \mathbf{H}^t \mathbf{v}, \mathbf{v} \rangle, \tag{10}$$

where $\mathbf{H}^t$ is the full Hessian $\nabla^2 F(\mathbf{x}^t)$ or its estimation. One can show that MetaAlgorithm 7 finds an $\mathcal{O}(\epsilon)$-SOSP w.p. at least $(1 - \delta)$ in $\tilde{\mathcal{O}}(1/\epsilon^{1.5})$ iterations. We defer the detailed analysis to Appendix C.

## 6.2 HESSIAN-FREE IMPLEMENTATION

Based on MetaAlgorithm 7, we propose a Hessian-free method named STR_free, which is summarized in Algorithm 8. STR_free leverages the full/stochastic Hessian and Estimator 4 to construct $\mathbf{H}^k$ and $\mathbf{g}^k$, respectively. Besides, it uses Lanczos method (Gould et al., 1999; Carmon & Duchi, 2018) as the QCQP solver, which can be implemented in a Hessian-free manner (i.e., using only Hessian-vector products without explicit Hessian matrix evaluations). Thus, $\mathbf{H}^k$ is only accessed through Hessian-vector products and is never explicitly constructed. Since Hessian-vector products can be computed in linear time (in terms of the dimension $d$) for many machine learning problems (Allen-Zhu, 2018b; Agarwal et al., 2017), Hessian-free methods are usually more practical than Hessian based ones for high dimensional problems. The following theorem, whose proof can be found in Appendix D, establishes the runtime complexity (i.e., the total complexity of stochastic gradient and Hessian-vector product evaluations (Zhou & Gu, 2019)) of STR_free.

---

**MetaAlgorithm 7** Inexact Trust Region Method II

---

**Input:** initial point $\mathbf{x}^0$, step size $r$, number of inner iterations $K$, constants $\delta, \zeta \in (0,1)$, $c_1$, $c_2$, number of outer iterations $T = \Theta(\log(1/\delta))$, sample size $s$ (optional)

1: **for** $t = 1$ **to** $T$ **do**
2:     $\mathbf{x}^t \leftarrow \text{INEXACTTR}_{\text{WEAK}}(\mathbf{x}^0, r, K, \zeta)$;
3:     Option I:         $\diamond$ high accuracy case (small $\epsilon$)
4:     $\mathbf{H}^t := \nabla^2 F(\mathbf{x}^t)$;
5:     Option II:        $\diamond$ low accuracy case (moderate $\epsilon$)
6:     Draw $s$ samples indexed by $\mathcal{H}$ and let $\mathbf{H}^t := \nabla^2 f(\mathbf{x}^t; \mathcal{H})$;
7:     Compute $\tilde{\mathbf{v}}^t$ by solving (10) up to accuracy $\sqrt{\epsilon L_2}$ with probability $1 - \delta/4$;
8:     **if** $\|\nabla F(\mathbf{x}^t)\| \le c_1 \epsilon$ and $\psi_t(\tilde{\mathbf{v}}^t) \ge -\sqrt{c_2 \epsilon L_2}$ **then**
9:         **return** $\mathbf{x}_\epsilon := \mathbf{x}^t$;
10:     **end if**
11: **end for**

12: **procedure** $\text{INEXACTTR}_{\text{WEAK}}(\mathbf{x}^0, r, K, \zeta)$
13:     **for** $k = 0$ **to** $K - 1$ **do**
14:         Compute $\mathbf{g}^k$ and $\mathbf{H}^k$ such that (9) holds with probability $1 - \frac{\zeta}{4K}$;
15:         Compute $\tilde{\mathbf{h}}^k$ by solving (8) up to accuracy $\epsilon^{1.5}/\sqrt{L_2}$ with probability $1 - \frac{\zeta}{4K}$;
16:         $\mathbf{x}^{k+1} := \mathbf{x}^k + \tilde{\mathbf{h}}^k$;
17:     **end for**
18:     Randomly select $\bar{k}$ from $\{0, \ldots, K - 1\}$;
19:     **return** $\mathbf{x}^{\bar{k}+1}$;
20: **end procedure**

---

**Algorithm 8** $\text{STR}_{\text{free}}$

---

1: In the same setting as MetaAlgorithm 7,
2:     construct gradient estimator $\mathbf{g}^k$ by Estimator 4;
3:     construct Hessian estimator $\mathbf{H}^k$ by
4:         Option I: $\mathbf{H}^k := \nabla^2 F(\mathbf{x}^k)$;
5:         Option II: Draw $s$ samples indexed by $\mathcal{H}$ and let $\mathbf{H}^k := \nabla^2 f(\mathbf{x}^k; \mathcal{H})$;
6:     use Lanczos method to solve QCQP subproblems.

---

**Theorem 6.1.** *Consider Algorithm 8 for solving problem (1). Let $\zeta = 1/3$, $r = \sqrt{\epsilon/L_2}$, $K = 4\sqrt{L_2}\Delta/\epsilon^{1.5}$, $T = \frac{3}{2}\log(2/\delta)$, $c_1 = 600$, $c_2 = 500$, and $s = \frac{32L_1^2}{\epsilon L_2}\log(4d/\delta)$. The hyper-parameters in Estimator 4 are set to the same values as those in Lemma 4.2. The number of iterations of Lanczos method is set to $\tilde{\mathcal{O}}(1/(L_2\epsilon)^{0.25})$. To find an $\mathcal{O}(\epsilon)$-SOSP w.p. at least $1 - \delta$, the runtime complexity is $\tilde{\mathcal{O}}(d \min\{n/\epsilon^{1.75}, 1/\epsilon^{2.75} + \sqrt{n}/\epsilon^2\}\log(1/\delta))$.*

To solve the QCQP more efficiently, here we develop a faster solver which is based on the AppxPCA method (Allen-Zhu & Li, 2016) and KatyushaX$^{\text{W}}$ (Allen-Zhu, 2018a). See details in Appendix E. By replacing Lanczos method with this solver in $\text{STR}_{\text{free}}$, we further improve the runtime complexity to $\tilde{\mathcal{O}}(d \min\{n/\epsilon^{1.5} + n^{0.75}/\epsilon^{1.75}, 1/\epsilon^{2.5} + \sqrt{n}/\epsilon^2\})$. We call this new algorithm $\text{STR}_{\text{free}}+$ whose details can be found in Appendix E. Table 2 shows that for the runtime complexities, both $\text{STR}_{\text{free}}$ and $\text{STR}_{\text{free}}+$ outperform existing methods. See more comparison and discussion in Appendix E.4.

## 7 EXPERIMENTS

Here we compare the proposed STR with several state-of-the-art (stochastic) cubic regularized algorithms and trust region approaches, including trust region (TR) algorithm (Conn et al., 2000), adaptive cubic regularization (ARC) (Cartis et al., 2011), sub-sampled cubic regularization (SCR) (Kohler & Lucchi, 2017a), stochastic variance-reduced cubic (SVRC) (Zhou et al., 2018c), Lite-SVRC (Zhou et al., 2018b), and SRVRC (Zhou & Gu, 2019). For STR, we estimate the gradient as the way in case (1). This is because such a method enjoys lower Hessian computational complexity over the way in case (2) and for most problems, computing their Hessian matrices is much more time-consuming than computing their gradients. For the subproblems in these compared methods, we use Lanczos

Table 2: Runtime complexities of STR$_{\text{free}}$, STR$_{\text{free}}$+, and other state-of-the-art methods.

| Algorithm | Runtime |
|---|---|
| Hessian-free Cubic (Carmon & Duchi, 2016) | $\tilde{\mathcal{O}}(\frac{dn}{\epsilon^2})$ |
| Fast-Cubic (Agarwal et al., 2017) | $\tilde{\mathcal{O}}(\frac{dn}{\epsilon^{1.5}} + \frac{dn^{0.75}}{\epsilon^{1.75}})$ |
| Stochastic Cubic (Tripuraneni et al., 2018) | $\tilde{\mathcal{O}}(\frac{d}{\epsilon^{3.5}})^*$ |
| SRVRC$_{\text{free}}$ (Zhou & Gu, 2019) | $\tilde{\mathcal{O}}(\min\{\frac{dn}{\epsilon^2}, \frac{d}{\epsilon^3}\})^*$ |
| STR$_{\text{free}}$ (this paper) | $\tilde{\mathcal{O}}(\min\{\frac{dn}{\epsilon^{1.75}}, \frac{d}{\epsilon^{2.75}} + \frac{dn^{0.5}}{\epsilon^2}\})$ |
| STR$_{\text{free}}$+ (this paper) | $\tilde{\mathcal{O}}(\min\{\frac{dn}{\epsilon^{1.5}} + \frac{dn^{0.75}}{\epsilon^{1.75}}, \frac{d}{\epsilon^{2.5}} + \frac{dn^{0.5}}{\epsilon^2}\})$ |

$^*$ These entries rely on an additional assumption: $\|\nabla f_i(\mathbf{x}) - \nabla F(\mathbf{x})\| \leq \sigma$ a.s.

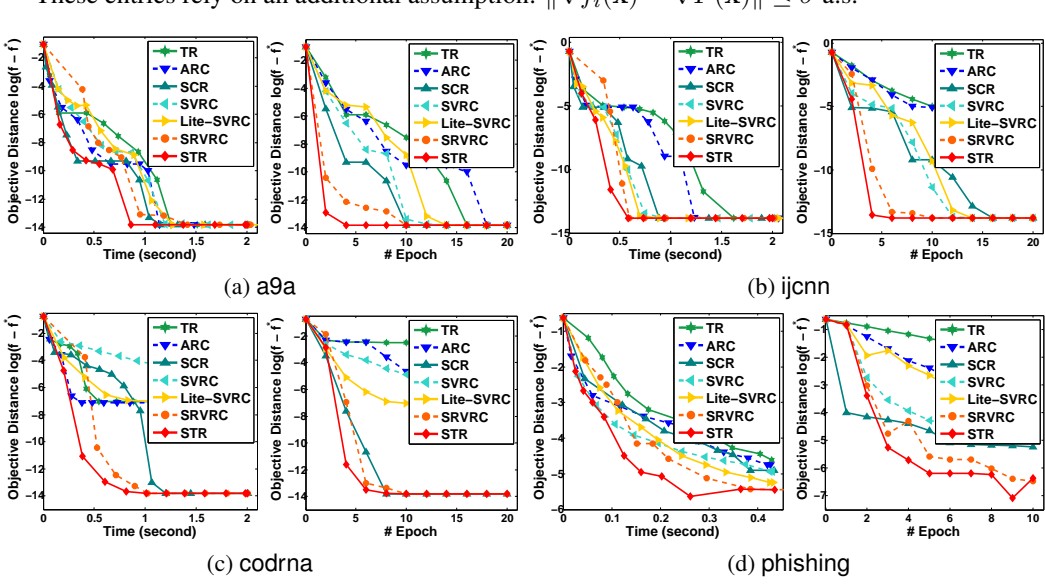

(a) a9a                        (b) ijcnn

(c) codrna                       (d) phishing

Figure 1: Comparison on the logistic regression with non-convex regularizer.

method (Gould et al., 1999; Kohler & Lucchi, 2017a) to solve the subproblem approximately in a Hessian-related Krylov subspace. We run simulations on seven datasets from LibSVM (a9a, ijcnn, codrna, phishing, w8a, epsilon and mnist). We run our algorithm for 40 epochs and use the output as the optimal value $f^*$ for sub-optimality estimation. Note the output has very small gradient already verified by Figure 2 and 4 in appendix. For all the considered algorithms, we set their initializations as zeros and tune their hyper-parameters optimally. For more experimental settings, e.g. details of testing datasets and algorithm parameter settings, please refer to Appendix F.

**Two evaluation non-convex problems.** Following (Kohler & Lucchi, 2017a; Zhou et al., 2018c), we evaluate all considered algorithms on two learning tasks: the logistic regression with non-convex regularizer and the nonlinear least square. Given $n$ data points $(\boldsymbol{x}_i, y_i)$ where $\boldsymbol{x}_i \in \mathbb{R}^d$ is the sample vector and $y_i \in \{-1, 1\}$ is the label, *logistic regression with non-convex regularizer* aims at distinguishing these two kinds of samples by solving the following problem $\min_{\boldsymbol{w}} \frac{1}{n} \sum_{i=1}^{n} \log(1 + \exp(-y_i \boldsymbol{w}^T \boldsymbol{x}_i)) + \lambda R(\boldsymbol{w}; \alpha)$, where the non-convex regularizer $R(\boldsymbol{w}; \alpha)$ is defined as $R(\boldsymbol{w}; \alpha) = \sum_{i=1}^{d} \alpha \boldsymbol{w}_i^2 / (1 + \alpha \boldsymbol{w}_i^2)$. The *nonlinear least square* problem fits the nonlinear data by minimizing $\min_{\boldsymbol{w}} \frac{1}{2n} \sum_{i=1}^{n} \left[ y_i - \phi(\boldsymbol{w}^T \boldsymbol{x}_i) \right]^2 + \lambda R(\boldsymbol{w}, \alpha)$. For these two kinds of problems, we set the parameters $\lambda = 10^{-3}$ and $\alpha = 10$ for all testing datasets.

**Comparison of Hessian based algorithms.** Figure 1 summarizes testing results on the non-convex logistic regression problem. For each dataset, we report the function value gap v.s. the overall algorithm running time which can reflect the overall computational complexity of an algorithm, and also show the function value gap v.s. Hessian sample complexity which reveals the complexity of Hessian computation. From Figure 1, one can observe that our proposed STR algorithm runs faster than the compared algorithms in terms of the algorithm running time, showing the overall superiority of STR. Furthermore, STR also reveals much sharper convergence curves in terms of the Hessian

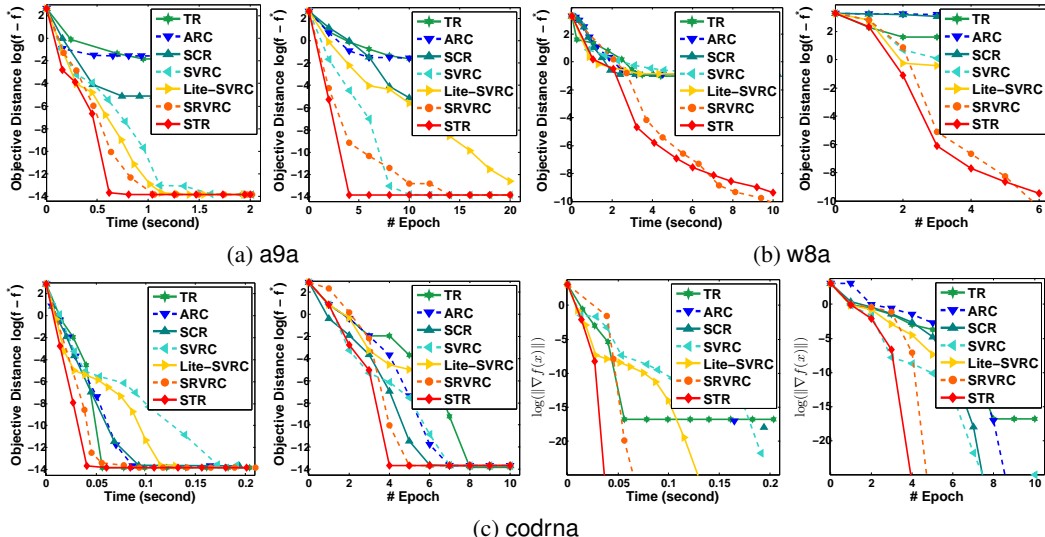

Figure 2: Comparison on the nonlinear least square problem.

sample complexity which is consistent with our theory. This is because to achieve an $\epsilon$-accuracy local minimum, the Hessian sample complexity of the proposed STR is $\tilde{\mathcal{O}}(n^{0.5}/\epsilon^{1.5})$ and is superior over the complexity of the compared methods (see the comparison in Sec. 4.2). Indeed, this also explains why our algorithm is also faster in terms of algorithm running time, since for most optimization problems, Hessian matrix is much more computationally expensive than the gradient and thus more efficient Hessian sample complexity means faster overall convergence speed. Note, as all compared methods need to compute the Hessian and gradient, their memory complexity are all $\mathcal{O}(d^2 + d)$.

Figure 2 displays results of the compared algorithms on the nonlinear least square problem. STR shows very similar behaviors as those in Figure 1. Specifically, STR achieves fastest convergence rate in terms of both algorithm running time and Hessian sample complexity. On the codrna dataset we further plot the gradient norm versus running time and Hessian sample complexity. One can obverse that the gradient in STR vanishes significantly faster than other algorithms which means that STR can find the stationary point with high efficiency. See Figure 4 in Appendix F.2 for more experimental results on gradient norm comparison. All these results confirm the superiority of the proposed STR.

**Comparison of Hessian-free algorithms.** Here we compare our proposed Hessian-free STR, namely STR$_{\text{free}}$, with other state-of-the-art Hessian-free algorithms on the two high-dimensional datasets, including epsilon and mnist (see details in Appendix F). Here we do not compare STR$_{\text{free}}$+, as it is based on AppxPCA method (Allen-Zhu & Li, 2016) and KatyushaX$^{\text{W}}$ (Allen-Zhu, 2018a) which require tuning a lot of hyper-parameters. From the results in Figure 3, one can observe that compared with other algorithms, our STR$_{\text{free}}$

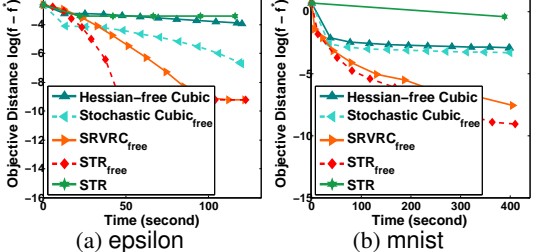

Figure 3: Comparison of Hessian-free algorithms on non-convex logistic and soft-max regressions.

achieves the best convergence speed which demonstrates its high efficiency in realistic applications. Besides, one also can find that STR$_{\text{free}}$ is much faster than Hessian based STR since computing full Hessian is actually much computationally expensive than the computation of the Hessian vector.

## 8 CONCLUSION

We proposed two stochastic trust region variants. Under two settings (whether stochastic first- and second-order oracle complexities are treated equally), the proposed methods achieve state-of-the-art oracle complexities. We also propose Hessian-free variants with lowest runtime complexity. Experimental results testify our theoretical implications and the efficiency of the proposed algorithms.

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

# A  APPENDIX

In this appendix, Sec. B first provides the proofs for the results in the manuscript. Then, we analyze MetaAlgorithm 7 and STR$_{\text{free}}$ in Sec. C and Sec. D, respectively. Next, in Sec. E, we develop a fast QCQP solver to further improve the computational complexity of STR$_{\text{free}}$. Finally, more experimental details and results are presented in Sec. F.

# B  DEFERRED PROOFS

## B.1  PROOF OF THEOREM 3.1

*Proof.* For simplicity of notation, we denote

$$\nabla_k \stackrel{\text{def}}{=} \nabla F(\mathbf{x}^k) - \mathbf{g}^k \text{ and } \nabla_k^2 \stackrel{\text{def}}{=} \nabla^2 F(\mathbf{x}^k) - \mathbf{H}^k.$$

From Assumption 2.3 we have

$$
\begin{aligned}
F(\mathbf{x}^{k+1}) \leq &F(\mathbf{x}^k) + \langle \nabla F(\mathbf{x}^k), \mathbf{h}^k \rangle + \frac{1}{2} \langle \nabla^2 F(\mathbf{x}^k) \mathbf{h}^k, \mathbf{h}^k \rangle + \frac{L_2}{6} \|\mathbf{h}^k\|^3 \\
= &F(\mathbf{x}^k) + \langle \nabla_k + \mathbf{g}^k, \mathbf{h}^k \rangle + \frac{1}{2} \langle [\nabla_k^2 + \mathbf{H}^k] \mathbf{h}^k, \mathbf{h}^k \rangle + \frac{L_2}{6} \|\mathbf{h}^k\|^3.
\end{aligned}
$$

Use the CauchySchwarz inequality to obtain

$$F(\mathbf{x}^{k+1}) \leq F(\mathbf{x}^k) + \langle \mathbf{g}^k, \mathbf{h}^k \rangle + \frac{1}{2} \langle \mathbf{H}^k \mathbf{h}^k, \mathbf{h}^k \rangle + \frac{L_2}{6} \|\mathbf{h}^k\|^3 + \|\nabla_k\| \|\mathbf{h}^k\| + \frac{1}{2} \|\nabla_k^2\| \|\mathbf{h}^k\|^2. \quad (11)$$

The requirement (9) together with the trust region radius $\|\mathbf{h}\| \leq r = \sqrt{\epsilon/L_2}$ allows us to bound

$$\|\nabla_k\| \|\mathbf{h}^k\| + \frac{1}{2} \|\nabla_k^2\| \|\mathbf{h}^k\|^2 \leq \frac{1}{3} \cdot \frac{\epsilon^{1.5}}{\sqrt{L_2}}. \quad (12)$$

The optimality of (5) indicates that there exists a dual variable $\lambda^k \geq 0$ so that (Corollary 7.2.2 in (Conn et al., 2000))

$$\text{First Order}: \mathbf{g}^k + \mathbf{H}^k \mathbf{h}^k + \frac{\lambda^k L_2}{2} \mathbf{h}^k = 0, \quad (13)$$

$$\text{Second Order}: \mathbf{H}^k + \frac{\lambda^k L_2}{2} \cdot \mathbf{I} \succcurlyeq 0, \quad (14)$$

$$\text{Complementary}: \lambda^k \cdot (\|\mathbf{h}^k\| - r) = 0. \quad (15)$$

Multiplying (13) by $\mathbf{h}^k$, we have

$$\langle \mathbf{g}^k + \mathbf{H}^k \mathbf{h}^k + \frac{\lambda^k L_2}{2} \mathbf{h}^k, \mathbf{h}^k \rangle = 0. \quad (16)$$

Additionally, using (14) we have

$$\langle (\mathbf{H}^k + \frac{\lambda^k L_2}{2} \mathbf{I}) \mathbf{h}^k, \mathbf{h}^k \rangle \geq 0,$$

which together with (16) gives

$$\langle \mathbf{g}^k, \mathbf{h}^k \rangle \leq 0. \quad (17)$$

Moreover, the complementary property (15) indicates $\|\mathbf{h}^k\| = \sqrt{\epsilon/L_2}$ as we have $\lambda^k > 3\sqrt{\epsilon/L_2} > 0$ before MetaAlgorithm 1 terminates. Plug (12), (16), and (17) into (11) and use $\|\mathbf{h}^k\| = \sqrt{\epsilon/L_2}$:

$$F(\mathbf{x}^{k+1}) \leq F(\mathbf{x}^k) - \frac{L_2 \lambda^k}{4} \cdot \frac{\epsilon}{L_2} + \frac{1}{2} \cdot \frac{\epsilon^{1.5}}{\sqrt{L_2}}. \quad (18)$$

Therefore, if we have $\lambda^k > 3\epsilon^{0.5}/\sqrt{L_2}$, then

$$F(\mathbf{x}^{k+1}) \leq F(\mathbf{x}^k) - \frac{1}{4\sqrt{L_2}} \cdot \epsilon^{1.5}. \quad (19)$$

Using Assumption 2.1, we find $\lambda^k \leq 3\epsilon^{0.5}/\sqrt{L_2}$ in no more than $4\sqrt{L_2} \cdot (F(\mathbf{x}^0) - F(\mathbf{x}^*))/\epsilon^{1.5}$ iterations.

We now show that once $\lambda^k \leq 3\epsilon^{0.5}/\sqrt{L_2}$, then $\mathbf{x}^{k+1}$ is already an $\mathcal{O}(\epsilon)$-SOSP: From (13), we have

$$\|\mathbf{g}^k + \mathbf{H}^k \mathbf{h}^k\| = \frac{L_2 \lambda^k}{2} \cdot \|\mathbf{h}^k\| \leq 2\epsilon. \tag{20}$$

The assumptions $\|\nabla_k\| \leq \epsilon/6$ and $\|\nabla_k^2\| \leq \sqrt{\epsilon L_2}/3$ together with the trust region radius $\|\mathbf{h}\| \leq \sqrt{\epsilon/L_2}$ imply

$$\|\nabla F(\mathbf{x}^k) + \nabla^2 F(\mathbf{x}^k)\mathbf{h}^k\| \leq \|\mathbf{g}^k + \mathbf{H}^k \mathbf{h}^k\| + \|\nabla_k\| + \|\nabla_k^2 \cdot \mathbf{h}^k\| \leq 2.5\epsilon. \tag{21}$$

On the other hand use Assumption 2.3 to bound

$$\|\nabla F(\mathbf{x}^{k+1}) - \nabla F(\mathbf{x}^k) - \nabla^2 F(\mathbf{x}^k)\mathbf{h}^k\| \leq \frac{L_2}{2}\|\mathbf{h}^k\|^2 \leq \frac{\epsilon}{2}.$$

Combining these two results gives $\|\nabla F(\mathbf{x}^{k+1})\| \leq 3\epsilon$.

Besides, using Assumption 2.3, $\|\nabla_k^2\| \leq \sqrt{\epsilon L_2}/3$, and (14), we derive the Hessian lower bound

$$\nabla^2 F(\mathbf{x}^{k+1}) \succcurlyeq \nabla^2 F(\mathbf{x}^k) - L_2 \cdot \|\mathbf{h}^k\|\mathbf{I} \succcurlyeq \mathbf{H}^k - \sqrt{\epsilon L_2}/3\mathbf{I} - L_2\|\mathbf{h}^k\|\mathbf{I} \succcurlyeq -\sqrt{12\epsilon L_2}\mathbf{I}.$$

Hence $\mathbf{x}^{k+1}$ is a $12\epsilon$-stationary point. Additionally, we have $\|\mathbf{h}^k\| = r$ according to the complementary condition (15) for all but the last iteration. $\qquad\square$

## B.2  PROOF OF LEMMA 4.1

*Proof.* Without loss of generality, we analyze the case $0 \leq k < q_2$ for ease of notation. We first focus on Option II. The proof for Option I follows the similar argument.

**Option II:** Define for $k = 0$ and $i \in [s_2']$

$$\mathbf{B}_i^0 \stackrel{\text{def}}{=} \nabla^2 f_i(\mathbf{x}^0) - \nabla^2 F(\mathbf{x}^0),$$

and define for $k \geq 1$ and $i \in [s_2]$

$$\mathbf{B}_i^k \stackrel{\text{def}}{=} \nabla^2 f_i(\mathbf{x}^k) - \nabla^2 f_i(\mathbf{x}^{k-1}) - (\nabla^2 F(\mathbf{x}^k) - \nabla^2 F(\mathbf{x}^{k-1})).$$

$\{\mathbf{B}_i^k\}$ is a martingale difference sequence. We have for all $k$ and $i$,

$$\mathbb{E}[\mathbf{B}_i^k | \mathbf{x}^k] = 0.$$

Besides, we use Assumption 2.2 for $k = 0$ to bound

$$\|\mathbf{B}_i^0\| \leq \|\nabla^2 f_i(\mathbf{x}^0)\| + \|\nabla^2 F(\mathbf{x}^0)\| = 2L_1, \tag{22}$$

and use Assumption 2.3 for $k \geq 1$ to bound

$$\|\mathbf{B}_i^k\| \leq \|\nabla^2 f_i(\mathbf{x}^k) - \nabla^2 f_i(\mathbf{x}^{k-1})\| + \|\nabla^2 F(\mathbf{x}^k) - \nabla^2 F(\mathbf{x}^{k-1})\| \leq 2\sqrt{\epsilon L_2}.$$

From the construction of $\mathbf{H}^k$, we have

$$\mathbf{H}^k - \nabla^2 F(\mathbf{x}^k) = \sum_{i=1}^{s_2'} \frac{\mathbf{B}_i^0}{s_2'} + \sum_{j=1}^{k} \sum_{i=1}^{s_2} \frac{\mathbf{B}_i^j}{s_2}.$$

Thus using the matrix Azuma's Inequality in Theorem 7.1 of (Tropp, 2012) and $k \leq p_2$, we have

$$Pr\{\|\mathbf{H}^k - \nabla^2 F(\mathbf{x}^k)\| \geq t\} \leq d \cdot \exp\{-\frac{t^2/8}{\sum_{i=1}^{s_2'} 4L_1^2/s_2'^2 + \sum_{j=1}^{k} \sum_{i=1}^{s_2} 4\epsilon L_2/s_2^2}\}$$

$$\leq d \cdot \exp\{-\frac{t^2/8}{4L_1^2/s_2' + 4p_2\epsilon L_2/s_2}\}.$$

Consequently, we have

$$Pr\{\|\mathbf{H}^k - \nabla^2 F(\mathbf{x}^k)\| \leq \sqrt{\epsilon L_2}\} \geq 1 - \delta/K_0.$$

by taking $t = \sqrt{\epsilon L_2}$, $s_2' = 16 L_1^2/(\epsilon L_2) \log(dK_0/\delta)$, $s_2 = 32 L_1/(\sqrt{\epsilon L_2}) \log(dK_0/\delta)$, and $p_2 = L_1/(2\sqrt{\epsilon L_2})$.

**Option I:** The proof is similar to the one of Option II except that we replace $\mathbf{B}_i^0$ with zero matrix. In such case, the matrix Azuma's Inequality implies

$$Pr\{\|\mathbf{H}^k - \nabla^2 F(\mathbf{x}^k)\| \geq t\} \leq d \cdot \exp\{-\frac{t^2/8}{\sum_{j=1}^k \sum_{i=1}^{s_2} 4\epsilon L_2/s_2^2}\} \leq d \cdot \exp\{-\frac{t^2/8}{4p_2\epsilon L_2/s_2}\}.$$

Thus by taking $t = \sqrt{\epsilon L_2}$, $s_2 = 32\sqrt{n} \log(dK_0/\delta)$, and $p_2 = \sqrt{n}$, we have the result.

**Amortized Complexity:** In option I, the choice of parameters ensures that: $s_2' \leq p_2 \times s_2$ and in option II: $n \leq p_2 \times s_2$. Consequently, the amortized stochastic second-order oracle complexity is bounded from above by $2s_2$. □

### B.3 PROOF OF LEMMA 4.2

Without loss of generality, we analyze the case $0 \leq k < q_1$ for ease of notation. Define for $k \geq 1$ and $i \in [s_1]$

$$\mathbf{a}_i^k \stackrel{\text{def}}{=} \nabla f_i(\mathbf{x}^k) - \nabla f_i(\mathbf{x}^{k-1}) - (\nabla F(\mathbf{x}^k) - \nabla F(\mathbf{x}^{k-1})).$$

$\{\mathbf{a}_i^k\}$ is a martingale difference sequence: for all $k$ and $i$

$$\mathbb{E}[\mathbf{a}_i^k | \mathbf{x}^k] = 0.$$

Besides, $\mathbf{a}_i^k$ has bounded norm:

$$\begin{aligned}
\|\mathbf{a}_i^k\| &\leq \|\nabla f_i(\mathbf{x}^k) - \nabla f_i(\mathbf{x}^{k-1})\| + \|\nabla F(\mathbf{x}^k) - \nabla F(\mathbf{x}^{k-1})\| \\
&\leq L_1 \|\mathbf{x}^k - \mathbf{x}^{k-1}\| + L_1 \|\mathbf{x}^k - \mathbf{x}^{k-1}\| \\
&\leq 2L_1 \sqrt{\epsilon/L_2}.
\end{aligned} \tag{23}$$

From the construction of $\mathbf{g}^k$, we have

$$\mathbf{g}^k - \nabla F(\mathbf{x}^k) = \sum_{j=1}^k \sum_{i=1}^{s_1} \frac{\mathbf{a}_i^j}{s_1}.$$

Recall the Azuma's Inequality. Using $k \leq p_1$, we have

$$\begin{aligned}
&Pr\{\|\mathbf{g}^k - \nabla F(\mathbf{x}^k)\| \geq t\} \\
&\leq \exp\{-\frac{t^2/8}{\sum_{j=1}^k \sum_{i=1}^{s_1} \frac{4\epsilon L_1^2}{L_2 s_1^2}}\} \leq \exp\{-\frac{t^2/8}{4\epsilon L_1^2 p_1/(s_1 L_2)}\}.
\end{aligned}$$

Take $t = \epsilon/6$ and denote $c = 1152$. To ensure that

$$Pr\{\|\mathbf{g}^k - \nabla F(\mathbf{x}^k)\| \geq \epsilon/6\} \leq \delta/K_0,$$

we need $\frac{cL_1^2}{L_2} \log \frac{K_0}{\delta} \leq \frac{\epsilon s_1}{p_1}$. The best amortized stochastic first-order oracle complexity can be obtained by solving the following two-dimensional programming:

$$\min_{p_1 \geq 1, s_1 \geq 1} \quad (n + s_1(p_1 - 1))/p_1$$

$$s.t. \quad \frac{cL_1^2}{L_2} \log \frac{K_0}{\delta} \leq \frac{\epsilon s_1}{p_1},$$

which has the solution $s_1 = \min\{n, \sqrt{\frac{n}{\epsilon} \cdot \frac{cL_1^2 \log \frac{K_0}{\delta}}{L_2}}\}$, and $p_1 = \max\{1, \sqrt{n\epsilon \cdot \frac{L_2}{cL_1^2 \log \frac{K_0}{\delta}}}\}$. Note that when we take $s_1 = n$, we directly compute $\mathbf{g}^k = \nabla F(\mathbf{x}^k)$ without sampling.

The amortized stochastic first-order oracle complexity is obtained by plugging in the choice of $s_1$ and $p_1$, which completes the proof.

### B.4 PROOF OF LEMMA 5.1

Without loss of generality, we analyze the case $0 \leq k < q_1$ for ease of notation. Define for $k \geq 1$ and $i \in [s_1]$

$$\mathbf{b}_i^k \stackrel{\text{def}}{=} \nabla f_i(\mathbf{x}^k) - \nabla f_i(\mathbf{x}^{k-1}) - \nabla^2 f_i(\tilde{\mathbf{x}})(\mathbf{x}^k - \mathbf{x}^{k-1})$$
$$- [\nabla F(\mathbf{x}^k) - \nabla F(\mathbf{x}^{k-1}) - \nabla^2 F(\tilde{\mathbf{x}})(\mathbf{x}^k - \mathbf{x}^{k-1})].$$

$\{\mathbf{b}_i^k\}$ is a martingale difference sequence: for all $k$ and $i$

$$\mathbb{E}[\mathbf{b}_i^k | \mathbf{x}^k] = 0.$$

Besides, $\mathbf{b}_i^k$ has bounded norm:

$$\|\mathbf{b}_i^k\| \leq \|\nabla f_i(\mathbf{x}^k) - \nabla f_i(\mathbf{x}^{k-1}) - \nabla^2 f_i(\tilde{\mathbf{x}})(\mathbf{x}^k - \mathbf{x}^{k-1})\|$$
$$+ \|\nabla F(\mathbf{x}^k) - \nabla F(\mathbf{x}^{k-1}) - \nabla^2 F(\tilde{\mathbf{x}})(\mathbf{x}^k - \mathbf{x}^{k-1})\|.$$

We can bound $\|\nabla f_i(\mathbf{x}^k) - \nabla f_i(\mathbf{x}^{k-1}) - \nabla^2 f_i(\tilde{\mathbf{x}})(\mathbf{x}^k - \mathbf{x}^{k-1})\|$ as follows.

$$\|\nabla f_i(\mathbf{x}^k) - \nabla f_i(\mathbf{x}^{k-1}) - \nabla^2 f_i(\tilde{\mathbf{x}})(\mathbf{x}^k - \mathbf{x}^{k-1})\|$$
$$= \|\int_0^1 \left[\nabla^2 f_i(\mathbf{x}^{k-1} + t(\mathbf{x}^k - \mathbf{x}^{k-1})) - \nabla^2 f_i(\tilde{\mathbf{x}})\right](\mathbf{x}^k - \mathbf{x}^{k-1})dt\|$$
$$\leq \int_0^1 L_2 \|t\mathbf{x}^k + (1-t)\mathbf{x}^{k-1} - \tilde{\mathbf{x}}\| dt \cdot \|\mathbf{x}^k - \mathbf{x}^{k-1}\|$$
$$\leq \int_0^1 \left(t\|\mathbf{x}^k - \tilde{\mathbf{x}}\| + (1-t)\|\mathbf{x}^{k-1} - \tilde{\mathbf{x}}\|\right)dt \cdot L_2 r$$
$$\leq L_2 k r^2,$$

where the first inequality follows from Assumption 2.3 and the last inequality holds because $\|\mathbf{x}^k - \tilde{\mathbf{x}}\| \leq kr$ and $\|\mathbf{x}^{k-1} - \tilde{\mathbf{x}}\| \leq kr$, where $r$ is the trust region radius. Similarly, we have $\|\nabla F(\mathbf{x}^k) - \nabla F(\mathbf{x}^{k-1}) - \nabla^2 F(\tilde{\mathbf{x}})(\mathbf{x}^k - \mathbf{x}^{k-1})\| \leq L_2 kr^2$. Thus, we bound

$$\|\mathbf{b}_i^k\| \leq 2L_2 kr^2 \leq 2p_1 \epsilon$$

From the construction of $\mathbf{g}^k$, we have

$$\mathbf{g}^k - \nabla F(\mathbf{x}^k) = \sum_{j=1}^k \sum_{i=1}^{s_1} \frac{\mathbf{b}_i^j}{s_1}.$$

We use $k \leq p_1$ and the Azuma's inequality to bound

$$Pr\{\|\mathbf{g}^k - \nabla F(\mathbf{x}^k)\| \geq t\}$$
$$\leq \exp\{-\frac{t^2/8}{\sum_{j=1}^k \sum_{i=1}^{s_1} \frac{4p_1^2\epsilon^2}{s_1^2}}\} \leq \exp\{-\frac{t^2/8}{4\epsilon^2 p_1^3/s_1}\}.$$

Thus, by taking $t = \epsilon/6$ and $c = 1152$, we need $\frac{s_1}{p_1^3} \geq c \log \frac{K_0}{\delta}$. Further we want $s_1 p_1 \simeq \mathcal{O}(n)$ and hence we take $p_1 = n^{0.25}$ and $s_1 = n^{0.75} c \log \frac{K_0}{\delta}$. The amortized stochastic first-order oracle complexity is bounded by $2s_1$.

## C ANALYSIS OF METAALGORITHM 7

We first show that INEXACTTR$_{\text{WEAK}}$ finds an $\mathcal{O}(\epsilon)$-SOSP in $\mathcal{O}(1/\epsilon^{1.5})$ iterations with probability at least $2/3$ as stated in the following lemma.

**Lemma C.1.** *Consider problem (1) under Assumptions 2.1-2.3. Suppose that the differential estimators $\mathbf{g}^k$ and $\mathbf{H}^k$ satisfy Eqn. (9) with probability at least $(1 - \frac{\zeta}{4K})$. Besides, suppose that $\tilde{\mathbf{h}}^k$ is an approximate solution to (8) such that w.p. $(1 - \frac{\zeta}{4K})$,*

$$\langle \mathbf{g}^k, \tilde{\mathbf{h}}^k \rangle + \frac{1}{2}\langle \mathbf{H}^k \tilde{\mathbf{h}}^k, \tilde{\mathbf{h}}^k \rangle \leq \langle \mathbf{g}^k, \mathbf{h}^k \rangle + \frac{1}{2}\langle \mathbf{H}^k \mathbf{h}^k, \mathbf{h}^k \rangle + \frac{\epsilon^{1.5}}{\sqrt{L_2}}, \quad (24)$$

*where $\mathbf{h}^k$ is a global solution to (8). By setting $\zeta = 1/3$, $r = \sqrt{\epsilon/L_2}$, and $K = 4\sqrt{L_2}\Delta/\epsilon^{1.5}$, INEXACTTR$_{\text{WEAK}}$ outputs a $500\epsilon$-SOSP w.p. at least $2/3$.*

*Proof.* Combining (11) and (24), we have w.p. $(1 - \frac{\zeta}{4K})$,

$$
\begin{aligned}
F(\mathbf{x}^{k+1}) \leq & \, F(\mathbf{x}^k) + \langle \mathbf{g}^k, \mathbf{h}^k \rangle + \frac{1}{2} \langle \mathbf{H}^k \mathbf{h}^k, \mathbf{h}^k \rangle + \frac{\epsilon^{1.5}}{\sqrt{L_2}} \\
& + \frac{L_2}{6} \|\tilde{\mathbf{h}}^k\|^3 + \|\nabla_k\| \|\tilde{\mathbf{h}}^k\| + \frac{1}{2} \|\nabla_k^2\| \|\tilde{\mathbf{h}}^k\|^2,
\end{aligned}
\tag{25}
$$

where $\mathbf{h}^k$ is a global solution to the QCQP (8) and $\tilde{\mathbf{h}}^k$ is an approximate solution satisfying (24). We let $\lambda^k$ denote the dual variable corresponding to the global solution $\mathbf{h}^k$ as defined in Lemma 2.1. We note that $\mathbf{h}^k$ and $\lambda^k$ are used only in our analysis. The INEXACT$_{\text{WEAK}}$ algorithm only requires the approximate solution $\tilde{\mathbf{x}}^k$ without knowledge of $\mathbf{h}^k$ or $\lambda^k$.

By the assumption that (9) holds with probability $(1 - \frac{\zeta}{4K})$ and the fact that $\|\tilde{\mathbf{h}}^k\| \leq r = \sqrt{L_2 \epsilon}$, we have w.p. $(1 - \frac{\zeta}{4K})$,

$$
\frac{L_2}{6} \|\tilde{\mathbf{h}}^k\|^3 + \|\nabla_k\| \|\tilde{\mathbf{h}}^k\| + \frac{1}{2} \|\nabla_k^2\| \|\tilde{\mathbf{h}}^k\|^2 \leq \frac{\epsilon^{1.5}}{2\sqrt{L_2}}.
\tag{26}
$$

Plugging (16), (17), and (26) into (25) and applying the union bound, we have w.p. at least $(1 - \frac{\zeta}{2K})$,

$$
F(\mathbf{x}^{k+1}) \leq F(\mathbf{x}^k) - \frac{L_2 \lambda^k \|\mathbf{h}^k\|^2}{4} + \frac{3\epsilon^{1.5}}{2\sqrt{L_2}} = F(\mathbf{x}^k) - \frac{L_2 \lambda^k r^2}{4} + \frac{3\epsilon^{1.5}}{2\sqrt{L_2}},
\tag{27}
$$

where the second inequality follows from (15):

$$
0 = \lambda^k (\|\mathbf{h}^k\| - r) = \lambda^k (\|\mathbf{h}^k\| - r)(\|\mathbf{h}^k\| + r) = \lambda^k (\|\mathbf{h}^k\|^2 - r^2).
\tag{28}
$$

Summing inequality (27) from $k = 0$ to $K - 1$ and applying the union bound, we have w.p. at least $(1 - \zeta/2)$,

$$
\frac{1}{K} \sum_{k=0}^{K-1} \lambda^k \leq \frac{4(F(\mathbf{x}^0) - F(\mathbf{x}^{K+1}))}{L_2 r^2 K} + \frac{6\epsilon^{1.5}}{L_2^{1.5} r^2} \leq \frac{4\Delta}{\epsilon K} + \frac{6\sqrt{\epsilon}}{\sqrt{L_2}},
\tag{29}
$$

where the second inequality follows from Assumption 2.1 and our choice of the trust region radius.

By sampling $\bar{k}$ uniformly from $\{0, \ldots, K - 1\}$, we obtain

$$
\mathbb{E}[\lambda^{\bar{k}}] = \frac{1}{K} \sum_{k=0}^{K-1} \lambda^k,
\tag{30}
$$

where the expectation is taken over the randomness of $\bar{k}$. Combining (29) and (30) and taking $K = 4\Delta\sqrt{L_2}/\epsilon^{1.5}$, we have w.p. at least $(1 - \zeta/2)$

$$
\mathbb{E}[\lambda^{\bar{k}}] \leq \frac{7\sqrt{\epsilon}}{\sqrt{L_2}}.
\tag{31}
$$

Since $\lambda^k$ is always no-negative, by Markov's inequality and the union bound, with probability at least $1 - \zeta$, we have

$$
\lambda^{\bar{k}} \leq \frac{14\sqrt{\epsilon}}{\zeta\sqrt{L_2}}.
\tag{32}
$$

By taking $\zeta = 1/3$, we have w.p. at least $2/3$, $\lambda^{\bar{k}} \leq 42\sqrt{\epsilon/L_2}$. The rest of the proof is similar to Theorem 3.1 and we have the result. $\qquad \square$

The following theorem shows that MetaAlgorithm 7 finds an $\mathcal{O}(\epsilon)$-SOSP w.p. $(1 - \delta)$ after running INEXACTTR$_{\text{WEAK}}$ for $\Theta(\log(1/\delta))$ times.

**Theorem C.1** (Iteration Complexity of MetaAlgorithm 7). *In the same setting as Lemma C.1, let $T = \frac{3}{2}\log(2/\delta)$, $c_1 = 600$, $c_2 = 500$, and $s = \frac{32L_1^2}{\epsilon L_2}\log(4d/\delta)$. Then MetaAlgorithm 7 finds a $600\epsilon$-SOSP with probability at least $(1 - \delta)$.*

*Proof.* By Lemma C.1 and our choice of $T$, with probability $(1 - \delta/2)$, at least one of $\mathbf{x}^t$ is a $500\epsilon$-SOSP. On the other hand, since $\psi_t(\tilde{\mathbf{v}}^t) \leq \psi_t(\mathbf{v}^t) + \sqrt{\epsilon L_2}$ with probability $1 - \delta/4$, if $\psi_t(\tilde{\mathbf{v}}^t) \geq -\sqrt{c_2 \epsilon L_2}$, then, w.p. $1 - \delta/4$,

$$\psi_t(\mathbf{v}^t) \geq \psi_t(\tilde{\mathbf{v}}^t) - \sqrt{\epsilon L_2} \geq -\sqrt{c_2 \epsilon L_2} - \sqrt{\epsilon L_2} \geq -\sqrt{550 \epsilon L_2}, \tag{33}$$

where the last inequality follows from our choice of $c_2$.

**Option I:** Since $\mathbf{H}^t = \nabla^2 F(\mathbf{x}^t)$ is the full Hessian, $\psi_t(\mathbf{v}^t)$ is the smallest eigenvalue of $\nabla^2 F(\mathbf{x}^t)$. Applying the union bound, we conclude that MetaAlgorithm 7 outputs a $600\epsilon$-SOSP w.p. $(1 - \delta)$.

**Option II:** Let $\mathbf{B}_i := \nabla^2 f_i(\mathbf{x}^t) - \nabla^2 F(\mathbf{x}^t)$ for $i \in \mathcal{H}$, then

$$\mathbf{H}^t - \nabla^2 F(\mathbf{x}^t) = \frac{1}{s} \sum_{i=1}^s \mathbf{B}_i. \tag{34}$$

By Assumption 2.2, we have

$$\|\mathbf{B}_i\| \leq \|\nabla^2 f_i(\mathbf{x}^t)\| + \|\nabla^2 F(\mathbf{x}^t)\| \leq 2L_1. \tag{35}$$

Applying the matrix Azuma's Inequality in Theorem 7.1 of Tropp (2012) leads to

$$Pr\{\|\mathbf{H}^t - \nabla^2 F(\mathbf{x}^t)\| \geq \sqrt{\epsilon L_2}\} \leq d \cdot \exp(\frac{-\epsilon L_2 s}{32 L_1^2}). \tag{36}$$

By taking $s = \frac{32 L_1^2}{\epsilon L_2} \log(4d/\delta)$ and applying the union bound, we have with probability $1 - \delta$,

$$\nabla^2 F(\mathbf{x}^t) \succcurlyeq \mathbf{H}^t - \sqrt{\epsilon L_2}\mathbf{I} \succcurlyeq (\psi_t(\mathbf{v}^t) - \sqrt{\epsilon L_2})\mathbf{I} \succcurlyeq -\sqrt{600 \epsilon L_2}\mathbf{I}, \tag{37}$$

where the last inequality follows from (33). This completes the proof. $\square$

## D    PROOF OF THEOREM 6.1

*Proof.* We first analyze the computational cost of Lanczos method. By Corollary 2 in (Carmon & Duchi, 2018), for any desired accuracy $\tilde{\epsilon}$, Lanczos method achieves this accuracy in $\mathcal{O}(\frac{r}{\sqrt{\tilde{\epsilon}}} \log \frac{r\sqrt{d}}{\tilde{\epsilon}p})$ Lanczos iterations w.p. at least $(1 - p)$. Without loss of generality, we assume that the number of Lanczos iterations is strictly smaller than the dimension $d$, otherwise the QCQP subproblem can be solved exactly. We note that each Lanczos iteration involves computation of one matrix-vector product. Therefore, to satisfy the condition (24) in Lemma C.1, one needs to evaluate $\tilde{\mathcal{O}}(1/(L_2\epsilon)^{0.25})$ Hessian-vector products of the form $\mathbf{H}^k\mathbf{v}$. Similarly, to solve (10) up to accuracy $\sqrt{\epsilon L_2}$ w.h.p., one needs to evaluate $\tilde{\mathcal{O}}(1/(L_2\epsilon)^{0.25})$ Hessian-vector products of the form $\mathbf{H}^t\mathbf{v}$.

In MetaAlgorithm 7, to verify whether the candidate solution $\mathbf{v}^t$ is indeed an $\mathcal{O}(\epsilon)$-SOSP, one needs at most $\mathcal{O}(n)$ stochastic gradient evaluations and $\mathcal{O}(\min\{n, \log(4d/\delta)L_1^2/(L_2\epsilon)\}/(L_2\epsilon)^{0.25})$ stochastic Hessian-vector product evaluations, where the latter one follows from the proof of Theorem 7. We proceed to analyze the computational complexity of the INEXACTTR$_{\text{WEAK}}$ procedure. Recall that the iteration complexity of MetaAlgorithm 7 is $\mathcal{O}(\log(1/\delta)/\epsilon^{1.5})$. Following Lemma 4.2 and Corollary 4.1, the stochastic first-order oracle complexity is $\tilde{\mathcal{O}}(\min\{n/\epsilon^{1.5}, \sqrt{n}/\epsilon^2\}\log(1/\delta))$. Following the proof of Lemma 4.1 and Corollary 4.1, when $p_2 = 1$, the overall stochastic Hessian sample complexity is $\tilde{\mathcal{O}}(\min\{n/\epsilon^{1.5}, 1/\epsilon^{2.5}\}\log(1/\delta))$. Since it takes $\tilde{\mathcal{O}}(1/\epsilon^{0.25})$ Lanczos iterations to meet the condition (24) as stated above, the overall stochastic Hessian-vector product oracle complexity is $\tilde{\mathcal{O}}(\min\{n/\epsilon^{1.75}, 1/\epsilon^{2.75}\}\log(1/\delta))$. Combining the stochastic first-order and Hessian-vector product complexities, the overall runtime is $\tilde{\mathcal{O}}(d \min\{n/\epsilon^{1.75}, 1/\epsilon^{2.75} + \sqrt{n}/\epsilon^2\}\log(1/\delta))$. $\square$

## E    A FASTER HESSIAN-VECTOR BASED QCQP SOLVER

We recall from the previous section that, to approximately solve a quadratic subproblem in STR$_{\text{free}}$, Lanczos method requires $\tilde{\mathcal{O}}(\min\{n/\epsilon^{0.25}, 1/\epsilon^{1.25}\})$ stochastic Hessian-vector product evaluations. In this section, we propose a faster QCQP solver with an $\tilde{\mathcal{O}}(\min\{n + n^{0.75}/\epsilon^{0.25}, 1/\epsilon\})$ complexity. Replacing Lanczos method with this QCQP solver in STR$_{\text{free}}$ results in a faster Hessian-free method, which we refer to as STR$_{\text{free}}$+.

## E.1 Convex Reformulation of QCQP

To begin with, we present a known result that is key to achieve faster algorithm than Lanczos method. We summarize this result in Lemma E.1 which shows that the trust region subproblem is equivalent to a convex QCQP.

**Lemma E.1.** *(Convex Reformulation of QCQP (Flippo & Jansen, 1996; Wang & Xia, 2017)) Denote $\lambda_{\min}$ as the smallest eigenvalue of $\mathbf{H}^k$. Let $\mathbf{u}_{\min}$ be a corresponding eigenvector. W.l.o.g., we assume that $\langle \mathbf{g}^k, \mathbf{u}_{\min} \rangle \leq 0$. Let $\mu = \min\{\lambda_{\min}, 0\}$. Then the QCQP (8) is equivalent to the convex problem*

$$\min_{\mathbf{h}\in\mathbb{R}^d, \|\mathbf{h}\|\leq r} q^k(\mathbf{h}) = \langle \mathbf{g}^k, \mathbf{h} \rangle + \frac{1}{2}\langle (\mathbf{H}^k - \mu I)\mathbf{h}, \mathbf{h} \rangle + \frac{1}{2}\mu r^2 \tag{38}$$

*in the sense that (8) and (38) have the same minimum function value. Moreover, when $\lambda_{\min} < 0$, for any optimal solution of (38), denoted by $\mathbf{h}_c^k$,*

$$\mathbf{h}_c^k + \frac{\sqrt{\langle \mathbf{h}_c^k, \mathbf{u}_{\min} \rangle^2 - \|\mathbf{u}_{\min}\|^2(\|\mathbf{h}_c^k\|^2 - r^2)} - \langle \mathbf{h}_c^k, \mathbf{u}_{\min} \rangle}{\|\mathbf{u}_{\min}\|^2}\mathbf{u}_{\min} \tag{39}$$

*is a global minimizer of the original QCQP (8).*

To perform the above reformulation, one needs to compute the exact eigenpair $(\lambda_{\min}, \mathbf{u}_{\min})$. Nevertheless, as we shall see, it is sufficient to compute an approximate eigenpair $(\tilde{\lambda}, \tilde{\mathbf{u}})$ such that

$$\lambda_{\min} \leq \tilde{\lambda} = \tilde{\mathbf{u}}^T \mathbf{H}^k \tilde{\mathbf{u}} \leq \lambda_{\min} + \tilde{\epsilon}, \tag{40}$$

where $\tilde{\epsilon}$ is a target accuracy to be determined later. We note that $\tilde{\epsilon} \leq 2L_2$ w.l.o.g. since $\|\mathbf{H}^k\| \leq L_2$. With this approximate eigenpair, it remains to solve the following convex problem

$$\min_{\mathbf{h}\in\mathbb{R}^d, \|\mathbf{h}\|\leq r} \tilde{q}^k(\mathbf{h}) = \langle \mathbf{g}^k, \mathbf{h} \rangle + \frac{1}{2}\langle (\mathbf{H}^k - \tilde{\mu}I)\mathbf{h}, \mathbf{h} \rangle + \frac{1}{2}\tilde{\mu}r^2 \tag{41}$$

where $\tilde{\mu} = \min\{0, \tilde{\lambda} - \tilde{\epsilon}\}$. One can check that the problem (41) well approximates (38).

**Corollary E.1.** *Let $q_*^k$ and $\tilde{q}_*^k$ be the minimum function value of (38) and (41), respectively. Assume $\lambda_{\min} \leq \tilde{\lambda} = \tilde{\mathbf{u}}^T \mathbf{H}^k \tilde{\mathbf{u}} \leq \lambda_{\min} + \tilde{\epsilon}$. Then*

$$|q_*^k - \tilde{q}_*^k| \leq \tilde{\epsilon}r^2. \tag{42}$$

We note that the above convex reformulation approach divides an indefinite QCQP into two subproblems: (i) computation of an approximate eigenpair $(\tilde{\lambda}, \tilde{\mathbf{u}})$; (ii) solving the convex problem (41). As we shall see, by exploiting the finite-sum structure of the Hessian $\mathbf{H}^k$, these two subproblems can be efficiently solved. We treat these two subproblems in the following two subsections, respectively.

## E.2 Finding the Smallest Eigenvector

To find a unit vector that satisfies requirement (40), we resort to the AppxPCA method (Allen-Zhu & Li, 2016), which first finds an approximate eigenvalue $\lambda = \lambda_{\min} - \tilde{\epsilon}$ via binary search and then applies Power method to the positive definite matrix $(\mathbf{H}^k - \lambda I)^{-1}$ for a logarithmic number of iterations. Computing $(\mathbf{H}^k - \lambda I)^{-1}\mathbf{v}$ for any vector $\mathbf{v}$ is equivalent to solving the $\tilde{\epsilon}$-strongly convex problem (Allen-Zhu & Li, 2018)

$$\min_{\mathbf{u}} \phi^k(\mathbf{u}) := \frac{1}{2}\mathbf{u}^T(\mathbf{H}^k - \lambda I)\mathbf{u} - \langle \mathbf{v}, \mathbf{u} \rangle \tag{43}$$

We note that $\mathbf{H}^k = \frac{1}{|\mathcal{S}|}\sum_{i\in\mathcal{S}} \nabla^2 f_i(\mathbf{x}^k)$. Specifically, in $\text{STR}_{\text{free}}$, either $|\mathcal{S}| = n$ (i.e., $\mathbf{H}^k$ is the full Hessian) or $|\mathcal{S}| = \tilde{\mathcal{O}}(L_1^2/(L_2\epsilon))$ by Lemma 4.1. Therefore, $\phi^k(\cdot)$ can be expressed as sum of non-convex functions

$$\phi^k(\mathbf{u}) = \frac{1}{|\mathcal{S}|}\sum_{i\in\mathcal{S}} \phi_i^k(\mathbf{u}) = \frac{1}{|\mathcal{S}|}\sum_{i\in\mathcal{S}} \left(\frac{1}{2}\mathbf{u}^T(\nabla^2 f_i(\mathbf{x}^k) - \lambda I)\mathbf{u} - \langle \mathbf{v}, \mathbf{u} \rangle\right). \tag{44}$$

By observing that each $\phi_i^k$ is non-convex and has $(4L_2)$-Lipschitz gradient, we can use KatyushaX$^S$ (Allen-Zhu, 2018a) to solve problem (43) in $\tilde{\mathcal{O}}(|\mathcal{S}| + |\mathcal{S}|^{3/4}\sqrt{L_2/\tilde{\epsilon}})$ stochastic Hessian-vector product (i.e., $\nabla^2 f_i(\mathbf{x}^k)\mathbf{u}$) evaluations. The following result is taken from (Agarwal et al., 2017, Section G.3), which gives the overall computation complexity of AppxPCA.

---

**Algorithm 9** Fast QCQP Solver

---

**Input:** $\mathbf{H}^k$, $\mathbf{g}^k$, $r$, $\tilde{\epsilon}$, $\tilde{\epsilon}_1$

1: Use AppxPCA to find $(\tilde{\lambda}, \tilde{\mathbf{u}})$ satisfying (40), in which the matrix inverse is solved by KatyushaX$^{\mathsf{S}}$;
2: Use KatyushaX$^{\mathsf{W}}$ to solve (41) up to accuracy $\tilde{\epsilon}_1$, i.e., find a vector $\tilde{\mathbf{h}}$ such that $\tilde{q}^k(\mathbf{h}) - \tilde{q}_*^k \leq \tilde{\epsilon}_1$ with high probability;
3: Return $\tilde{\mathbf{h}} + (\sqrt{\langle\tilde{\mathbf{h}}, \tilde{\mathbf{u}}\rangle^2 - \|\tilde{\mathbf{u}}\|^2(\|\tilde{\mathbf{h}}\|^2 - r^2)} - \langle\tilde{\mathbf{h}}, \tilde{\mathbf{u}}\rangle)\tilde{\mathbf{u}}/\|\tilde{\mathbf{u}}\|^2$.

---

**Algorithm 10** STR$_{\text{free}}$+

---

1: In the same setting as MetaAlgorithm 7,
2:     construct gradient estimator $\mathbf{g}^k$ by Estimator 4;
3:     construct Hessian estimator $\mathbf{H}^k$ by
4:       Option I: $\mathbf{H}^k := \nabla^2 F(\mathbf{x}^k)$;
5:       Option II: Draw $s$ samples indexed by $\mathcal{H}$ and let $\mathbf{H}^k := \nabla^2 f(\mathbf{x}^k; \mathcal{H})$;
6:     use Algorithm 9 to solve QCQP subproblems.

---

**Lemma E.2.** *Let $\mathbf{H}^k = \frac{1}{|\mathcal{S}|}\sum_{i\in\mathcal{S}} \nabla^2 f_i(\mathbf{x}^k) \in \mathbb{R}^{d\times d}$, where $\|\nabla^2 f_i(\mathbf{x}^k)\| \leq L_2$. With probability at least $1 - p$, AppxPCA produces a unit vector $\mathbf{u}$ satisfying $\mathbf{u}^T\mathbf{H}^k\mathbf{u} \leq \lambda_{\min} + \tilde{\epsilon}$. The total stochastic Hessian-vector product oracle complexity is $\tilde{\mathcal{O}}(|\mathcal{S}| + |\mathcal{S}|^{3/4}\sqrt{L_2/\tilde{\epsilon}})$.*

### E.3 SOLVING THE CONVEX QCQP

In what follows, we show that the convex problem (41) can be solved efficiently. We first observe that problem (41) has a finite-sum structure and can be rewritten as an unconstrained problem of the form

$$\min_{\mathbf{h}\in\mathbb{R}^d} \frac{1}{|\mathcal{S}|}\sum_{i\in\mathcal{S}} \tilde{q}_i^k(\mathbf{h}) + \Psi(\mathbf{h}) = \frac{1}{|\mathcal{S}|}\sum_{i\in\mathcal{S}}\left(\langle\mathbf{g}^k, \mathbf{h}\rangle + \frac{1}{2}\langle(\nabla^2 f_i(\mathbf{x}^k) - \tilde{\mu}I)\mathbf{h}, \mathbf{h}\rangle + \frac{1}{2}\tilde{\mu}r^2\right) + \Psi(\mathbf{h}), \quad (45)$$

where $\Psi(\mathbf{h}) = 0$ if $\|\mathbf{h}\| \leq r$, otherwise $\Psi(\mathbf{h}) = +\infty$. We note that each $\tilde{q}_i^k(\mathbf{h})$ in (45) has $(4L_2)$-Lipschitz continuous gradient since $\|\nabla^2 f_i(\mathbf{x}^k)\| \leq L_2$ and $\tilde{\epsilon} \leq 2L_2$. Therefore, we can use KatyushaX$^{\mathsf{W}}$ (Allen-Zhu, 2018a) to solve (45). By (Allen-Zhu, 2018a, Theorem 4.6), KatyushaX$^{\mathsf{W}}$ finds a point $\mathbf{h}$ such that $\mathbb{E}[\tilde{q}^k(\mathbf{h}) - \tilde{q}_*^k] \leq \tilde{\epsilon}_1$ using $\tilde{\mathcal{O}}(|\mathcal{S}| + |\mathcal{S}|^{3/4}\sqrt{L_2} \cdot r/\sqrt{\tilde{\epsilon}_1})$ stochastic Hessian-vector products, where $\tilde{\epsilon}_1$ is the target accuracy to be determined later.

### E.4 PUTTING IT ALL TOGETHER

The complete procedure of our fast QCQP solver is summarized in Algorithm 9. Combining all the above results and setting $r = \sqrt{\epsilon/L_2}$, $\tilde{\epsilon} = \sqrt{\epsilon L_2}/2$, and $\tilde{\epsilon}_1 = \tilde{\epsilon}r^2$, one can find an approximate solution to QCQP (8) satisfying requirement (24) in $\tilde{\mathcal{O}}(|\mathcal{S}| + |\mathcal{S}|^{3/4}L_2^{0.25}/\epsilon^{0.25})$ stochastic Hessian-vector product evaluations. By replacing Lanczos method with this solver in STR$_{\text{free}}$, we derive a new Hessian-free method called STR$_{\text{free}}$+, which is summarized in Algorithm 10. The following theorem establishes the overall runtime complexity of STR$_{\text{free}}$+ for finding an $\epsilon$-SOSP.

**Theorem E.1.** *Consider Algorithm 10 for solving problem (1). Let $\zeta = 1/3$, $r = \sqrt{\epsilon/L_2}$, $K = 4\sqrt{L_2}\Delta/\epsilon^{1.5}$, $T = \frac{3}{2}\log(2/\delta)$, $c_1 = 600$, $c_2 = 500$, and $s = \frac{32L_1^2}{\epsilon L_2}\log(4d/\delta)$. The hyper-parameters in Estimator 4 are set to the same values as those in Lemma 4.2. Besides, let $\tilde{\epsilon} = \sqrt{\epsilon L_2}/2$ and $\tilde{\epsilon}_1 = \tilde{\epsilon}r^2$ in Algorithm 9. To find an $\mathcal{O}(\epsilon)$-SOSP w.p. at least $1 - \delta$, the runtime complexity is $\tilde{\mathcal{O}}(d\min\{n/\epsilon^{1.5} + n^{0.75}/\epsilon^{1.75}, 1/\epsilon^{2.5} + \sqrt{n}/\epsilon^2\}\log(1/\delta))$.*

*Proof.* The proof directly follows from that in Sec. D. $\qquad\square$

We compare the runtime complexity of STR$_{\text{free}}$ and STR$_{\text{free}}$+ with existing Hessian free methods in Table 2. One can see that STR$_{\text{free}}$ strictly outperforms Hessian-free Cubic. Besides, STR$_{\text{free}}$ outperforms Fast-Cubic if $n \geq \Omega(1/\epsilon^{4/3})$, which is a mild condition for large-scale problems in the moderate accuracy case. STR$_{\text{free}}$+ strictly outperforms both Hessian-free Cubic and Fast-Cubic. We

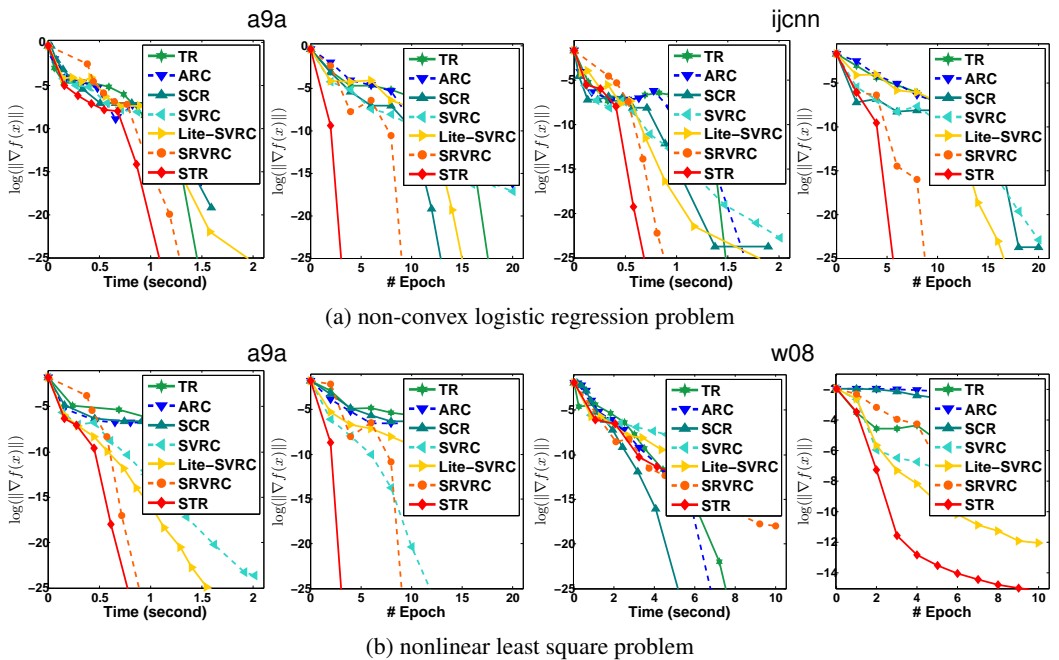

(a) non-convex logistic regression problem

(b) nonlinear least square problem

Figure 4: Comparison of gradient norm on both the non-convex logistic regression and nonlinear least square problems.

note that the runtime analyses in (Tripuraneni et al., 2018; Zhou & Gu, 2019) rely on an additional assumption which states that for all $\mathbf{x}$, with probability 1,

$$\|\nabla f_i(\mathbf{x}) - \nabla F(\mathbf{x})\| \leq \sigma. \tag{46}$$

Under this additional assumption, one can use the same argument as in Section B.2 and B.3 to prove that $\text{STR}_{\text{free}}$ achieves a runtime complexity of $\tilde{\mathcal{O}}(d \min\{n/\epsilon^{1.75}, n^{0.5}/\epsilon^2 + 1/\epsilon^{2.75}, 1/\epsilon^3\})$[1]. Similarly, the runtime complexity of $\text{STR}_{\text{free}}+$ would be $\tilde{\mathcal{O}}(d \min\{n/\epsilon^{1.5} + n^{0.75}/\epsilon^{1.75}, 1/\epsilon^{2.5} + \sqrt{n}/\epsilon^2, 1/\epsilon^3\})$. In this sense, both $\text{STR}_{\text{free}}$ and $\text{STR}_{\text{free}}+$ outperform Stochastic Cubic and $\text{SRVRC}_{\text{free}}$.

Table 3: Descriptions of the five testing datasets.

|          | #sample | #feature |          | #sample | #feature |
|----------|---------|----------|----------|---------|----------|
| a9a      | 32,561  | 123      | w8a      | 49,749  | 300      |
| ijcnn    | 49,990  | 22       | phishing | 7,604   | 68       |
| codrna   | 28,305  | 8        | mnist    | 60,000  | 784      |
| epsilon  | 40,000  | 2,000    |          |         |          |

# F    ADDITIONAL EXPERIMENTAL RESULTS

## F.1    MORE EXPERIMENTAL DETAILS

**Descriptions of Testing Datasets.** We briefly introduce the seven testing datasets in the manuscript. Among them, six datasets are provided by the LibSVM website[2], including (a9a, ijcnn, codrna, phishing, w8a and epsilon). The detailed information is summarized in Table 3. We can observe that these datasets are different from each other in feature dimension, training samples, etc.

---

[1]To obtain this complexity, one needs to replace the full gradient in line 2 of Estimator 4 (and line 6 of MetaAlgorithm 7) with a mini-batch stochastic gradient when $n \geq \Omega(1/\epsilon^2)$.

[2]https://www.csie.ntu.edu.tw/ cjlin/libsvmtools/datasets/

**Experimental Settings.** In the manuscript, following SVRC (Zhou et al., 2018c) and Lite-SVRC (Zhou et al., 2018b), we select hyper parameters from a set, namely $s_1$ from $\{0.2n, 0.6n, n\}$, $s_2$ from $\{0.01n, 0.1n, 0.2n\}$, $p_1$ and $p_2$ from $\{0.01n^{0.5}, 0.05n^{0.5}, 0.1n^{0.5}\}$. For the Hessian estimation at the beginning of each $p_2$ iterations, we use full Hessian. Similarly, for the gradient estimation at the beginning of each $p_1$ iterations, we adopt the full gradient as the gradient estimation.

**Memory Analysis.** SVRC, Lite-SVRC, and our method need to store the previous and current gradient and Hessian and thus their memory complexity is $2(d^2 + d)$. TR, CR (ARC) and SCR need to compute current and Hessian and thus has complexity $d^2 + d$. So these memory is of the same order but our method is much faster than TR, CR and SCR both validated by theory and experiments.

### F.2    MORE EXPERIMENTS

Here we give more experimental results on the gradient norm v.s. the algorithm running time and the Hessian sample complexity. Due to the space limit, in the manuscript we only provide the gradient-norm related results on the codrna dataset. Here we provide results of a9a and ijcnn datasets in Figure 4. One can observe that on both the logistic regression with non-convex regularizer and the nonlinear least square problems, the proposed algorithm always shows sharper convergence behavior in terms of both the running time and the Hessian sample complexity. These observations are consistent with the results in Figure 2 in the manuscript. All these results demonstrate the high efficiency of our proposed algorithm and also confirm our theoretical implication.

