# OpenReview forum: "A Stochastic Trust Region Method for Non-convex Minimization"
_ICLR.cc/2020/Conference — Reject_

### Official Review · AnonReviewer3 · 2019-10-23
**Official Blind Review #3**

**Rating:** 3

**Review:**

In this paper, the authors improved the state-of-the-art result by using inexact gradient and Hessian estimation in the training and proving under this case, it still have good performance. In order to control the difference between gradient and approximation gradient, the author use variance reduced estimator to exploit the correlation between consecutive iterations. In addition, the author consider for two cases with the importance of second-order oracle and use Hessian to improve the approximation of gradient when first-order and second-order oracle are equally important. Furthermore, the authors proposed a refined algorithm in practice which only uses stochastic gradient and Hessian-vector product information and showed better experiment results.
In general, the paper is well written and easy to follow, but I still have some questions about this paper.
First, there lacks explanation about why using biased estimators can still guarantee the convergence. For estimator 3 and estimator 4, it seems like that the stochastic gradient and Hessian are not unbiased approximation of the true value. That is a bit strange, since most popular stochastic estimators including stochastic gradient and stochastic variance-reduced gradient are unbiased approximation, which could guarantee that when the number of samples is large enough, the estimators can approximate the true quantities very well.  More discussion about this issue is needed. .
Second, to the best of my knowledge, the trust region radius is changing during the iteration for basic trust region algorithm. However, in meta-algorithm 1, the radius is fixed to a very small quantity which is related to the accuracy \epsilon. I am very surprised that with a fixed small radius, STR can still achieve the best result among all baseline algorithms, especially compared with trust region algorithm with adaptive radius. Can the authors explain this phenomenon?
Third, there is no discussion on space complexity. In practice, it is important to consider the space complexity. However, this work did not provide any space complexity analysis, especially to compare with first-order algorithms. It is interesting to see the trade-off between space complexity and time complexity among both first-order algorithms and second-order algorithms.
Fourth, the results of Lemma 4.1 and 4.2 are all in high probability. When the probability $\delta$ is small, then the number of sample is large. Thus, it is not fair that the authors simply ignored them when they compared their algorithms with deterministic algorithm. Furthermore, in practice, to choose small $\delta$ may cause the total number of samples very big. The authors need to make more comments on this issue.
Finally, I think the experiment results contradict the theoretical results. From figure 1, it can be seen that the STR method including many other baseline algorithms converge to the global minimum with linear convergence speed (a9a, epoch). However, the convergence analysis provided by the authors claim that the convergence speed is sublinear. I believe such a difference is due to a fact that the initialized parameter is near to the global minimum, thus the optimization landscape here is actually convex but not non-convex. That makes the experiment results vacuous.


**Experience Assessment:**

I have read many papers in this area.

**Review Assessment: Checking Correctness Of Derivations And Theory:**

I assessed the sensibility of the derivations and theory.

**Review Assessment: Checking Correctness Of Experiments:**

I assessed the sensibility of the experiments.

**Review Assessment: Thoroughness In Paper Reading:**

I read the paper at least twice and used my best judgement in assessing the paper.

---

> ### Author Response · Authors · 2019-11-12
> **Response to reviewer's comments.**
>
> 1. biased estimator:
> Estimators 3 and 4 are indeed unbiased gradient and Hessian estimators, respectively, which can be checked via a simple induction. Taking the gradient estimator as an example. Clearly $g^{0}$ is unbiased. Suppose $E[g^{k-1}]=\nabla F(x^{k-1})$ is unbiased. Then we have $E[g^{k}] = E[g^{k-1}] + E[\nabla f(x^{k}; G) - \nabla f(x^{k-1}; G)] = \nabla F(x^{k-1}) + \nabla F(x^{k}) - \nabla F(x^{k-1}) = \nabla F(x^{k})$, meaning the gradient estimation is unbiased.
>
> 2. trust region size:
> There are two reasons that our variance-reduced STR algorithms outperform trust region (TR) methods. First, the faster speed of our STR contributes to its gradient and Hessian estimators. These two estimators only require a small minibatch of samples to estimate very accurate gradient and Hessian. In contrast, TR needs to access the whole dataset which is time-consuming, especially for large-scale datasets. So our STR has much lower algorithm running time and Hessian sample complexity over TR at each iteration. Moreover, in the experiments, we use a loose TR size to achieve satisfactory performance. This is because a loose TR size constraint allows the algorithm to select a more proper stepsize for fast decreasing the objective according to the current landscape of the problem. Compared with adaptive stepsize in TR which computes the objective several times with substantial computational time especially for high-dimensional problems, this fixed stepsize strategy may not be as good as the adaptive stepsize strategy but it does not increase any computational cost. Since for second-order algorithms, computing their Hessian and gradient dominates the main computational cost, our STR outperforms TR method.
>
> 3. space complexity
> For non-Hessian-free algorithms in Table 1, such as TR, CR and our STR, they all need to explicitly compute the Hessian matrix. So their memory cost is $O(d^2)$. To alleviate such cost, we propose a Hessian-free variant, i.e. STR_{free} in Algorithm 8, which only requires Hessian-vector products. So the memory cost is $O(d)$. For this, we have discussed our algorithms and other baselines in the third paragraph in Sec. 7.
>
> 4. high probability convergence
> We acknowledge that a very small failure probability $\delta$ would introduce a marginal cost ($log 1/\delta$) to the overall complexity. However, the dependence on the failure probability $\delta$ is ubiquitous in all stochastic algorithms with high probability convergence. Importantly, our results compared to deterministic methods enjoy a polynomial reduction to the dependence on the number $n$ of component function, at the cost of a minor logarithmic dependence on $1/\delta$. For example, the $O(n/\epsilon^{1.5})$ SSO complexity in deterministic Trust Region is reduced to $O(\log(1/\delta)*\sqrt{n}/\epsilon^{1.5})$ in our Stochastic Trust Region.
>
> 5. experiment
> Thanks. Our experiments follow the conventional settings in (Kohler et al, ICML'17; Zhou et al, ICML'18), including the optimization problems and parameter settings. One can observe the same phenomenon in these papers. In all experiments, the initialization for all methods are the origin and hence are not close to a minimum. After receiving the comment, we carefully checked the optimization problems on the testing datasets and found that the minimum eigenvalue of the Hessian matrix on the ijcnn testing dataset is negative but not very small (at the order of -1e-4). Other than that, the problems on other datasets, e.g. phishing and w08, are strongly non-convex. So the algorithm's behaviors are a little different.

---

### Official Review · AnonReviewer4 · 2019-10-24
**Official Blind Review #4**

**Rating:** 6

**Review:**

This paper proposes new stochastic trust region algorithms for non-convex finite-sum minimization problems. The first algorithm STR1 has lower second order oracle complexity, while STR2 has lower first order + second order oracle complexity. The authors also give a Hessian-free implementation of stochastic trust region algorithm. Technically, the authors first analyze trust region methods with inexact gradient and Hessian estimation, and then implement efficient gradient and Hessian estimators.

Overall this paper is well-written and easy to follow. I would recommend acceptance.

**Experience Assessment:**

I have read many papers in this area.

**Review Assessment: Checking Correctness Of Derivations And Theory:**

I assessed the sensibility of the derivations and theory.

**Review Assessment: Checking Correctness Of Experiments:**

I assessed the sensibility of the experiments.

**Review Assessment: Thoroughness In Paper Reading:**

I read the paper at least twice and used my best judgement in assessing the paper.

---

> ### Author Response · Authors · 2019-11-14
> **Response to reviewer's comments.**
>
> We thank the reviewer for the positive comments.

---

### Official Review · AnonReviewer1 · 2019-10-24
**Official Blind Review #1**

**Rating:** 8

**Review:**

The authors propose a new analysis for trust region methods with approximate models. Using this result, they propose a number of methods to create stochastic trust region methods by constructing approximate quadratic models (based on a stochastic first and second order estimate) which satisfy the requirements for convergence. The effectiveness of the derived methods is evaluated empirically on two standard non-convex regression problems.

This paper is overall an interesting contribution which proposes a number of competitive methods for achieving approximate local minima in a stochastic regime, with both hessian based and “hessian-free” methods. A couple of minor points:
- In the experiments, it would be helpful to also include some measure of uncertainty (such as standard error bars) in the plots given the stochastic nature of the problem (although I do not expect high variance given the construction of the algorithm).
- It would be helpful to indicate which results still hold in the online setting (not finite sum). Indeed, from the proof of theorem 3.1, MetaAlgorithm 1 does not seem to rely on the finite sum setting, which is mostly used for analyzing the variance-reduced estimators. This would be helpful as it would enable MetaAlgorithm 1 to be used with appropriate variance-reduced estimators in settings beyond the finite-sum problems.


**Experience Assessment:**

I do not know much about this area.

**Review Assessment: Checking Correctness Of Derivations And Theory:**

I did not assess the derivations or theory.

**Review Assessment: Checking Correctness Of Experiments:**

I assessed the sensibility of the experiments.

**Review Assessment: Thoroughness In Paper Reading:**

I made a quick assessment of this paper.

---

> ### Author Response · Authors · 2019-11-12
> **Response to reviewer's comments.**
>
> 1. measure of uncertainty in the experiment
> Thanks. We will add standard error bars in our revision.
>
> 2. results under the online setting
> Thanks for your suggestions. Yes, the proposed algorithms are applicable to the online setting. Under this setting, we can obtain very similar results as finite-sum setting. Specifically, to achieve our results under finite-sum setting, we only need to set proper gradient and Hessian minibatch sizes at each iteration such that the estimation errors of gradient and Hessian estimations are small, (see Eqn. (9)). For online setting, we also only need to satisfy Eqn. (9) which like finite-sum setting, can be achieved by setting proper gradient and Hessian minibatch sizes. We will discuss this and include the detailed results in our revision.

---

### Official Review · AnonReviewer2 · 2019-11-03
**Official Blind Review #2**

**Rating:** 3

**Review:**

This paper applies the spider algorithm (Fang et al., 2018) for reducing variance in first-order stochastic optimization to a second order optimization algorithm, i.e., trust region algorithm.
Due to the new gradient and Hessian estimators in Fang et al., 2018, the proposed stochastic trust region algorithms in this paper achieve $O(\min\{1/\epsilon^2,\sqrt{n}/\epsilon^{1.5}\})$ stochastic second-order oracle (SSO) complexity. This result improves the SSO of stochastic variance-reduced cubic (SVRC) by a factor of $n^{1/6}$. This paper has a moderate contribution because of the new algorithms and improved complexity results. However, the idea of variance reduction is not novel and the result is not surprising since the improvement comes purely from spider (Fang et al., 2018) and thus this work is somewhat incremental. The paper is in general fairly written without many typos or unclear statements. But some places are verbose and necessarily complicated. Moreover, the comparison between this paper and existing work is not clear and straightforward. Lastly, the presentation of the current paper is unnecessarily verbose and complicated.

The comparison of this paper with a similar paper by Zhou & Gu (2019) is not convincing. In the remark after Corollary 4.1 and in Section 6.1, the authors mentioned that the SRVRC algorithm proposed by Zhou & Gu (2019) achieves similar complexity as this paper. But the authors did not present the complexity of SRVRC in Table 1. This is not appropriate since it is very similar and related to this paper. The authors should compare with existing work in a more clear and fair way.

In Remark 3.1 and the discussion after that, the authors argued that the exact step size control is crucial to the sample efficiency of stochastic trust region algorithms. However, the cubic regularization based algorithm in Zhou & Gu (2019) can also achieve the same second order oracle complexity. Therefore, I don’t think the arguments in the two paragraphs after Remark 3.1 are the key reason for the improvement. Instead, the spider estimator that greatly reduces the variance is the key point leading to the improved sample efficiency.

I find the presentation of this paper is very verbose and unnecessarily long (10 pages, 8 algorithm boxes and 8 theorems/lemmas in the main paper). Many places can be simplified or combined in order to increase the readability of the main theorems. Some intermediate results may also be moved to the appendix if necessary.

Algorithm 2 and 3 are almost identical since the gradient and Hessian estimators ($g^k, H^k$) are represented in the same form. I don’t see the point of repeating the algorithms twice. In the Estimator 3, the two options can be simply combined by setting $s_2’=\min\{1/\epsilon, n\}$ since according to Lemma 4.1 $s_2’=1/\epsilon$ in option II.

The paper talks about $\epsilon$-SOSP, $\tilde{O}(\epsilon)$-SOSP, $12\epsilon$-SOSP and so on in many places. It would be better to be consistent and use the same notation.

In the text after Corollary 4.1, there are some typos in the complexities where a $\min$ operator is missing. Moreover, the complexity of Zhou & Gu (2019) presented here seems not correct. I quickly checked their paper and found in their table that the SFO complexity of Zhou & Gu (2019) is $\tilde{O}(\min\{n/\epsilon^{3/2},n^{1/2}/\epsilon^2,1/\epsilon^3\})$, which is in fact smaller than the complexity of STR1 in this paper. Therefore, I think the comparison in this paragraph is not correct.


**Experience Assessment:**

I have published one or two papers in this area.

**Review Assessment: Checking Correctness Of Derivations And Theory:**

I assessed the sensibility of the derivations and theory.

**Review Assessment: Checking Correctness Of Experiments:**

I carefully checked the experiments.

**Review Assessment: Thoroughness In Paper Reading:**

I read the paper at least twice and used my best judgement in assessing the paper.

---

> ### Author Response · Authors · 2019-11-12
> **Response to reviewer's comments. Please note that Zhou & Gu (2019) has updated their arXiv submission during the review period.**
>
> 1. Novelty:
> Our novelty is two-folded. Firstly, for the first time, we prove the vanilla trust region method (MetaAlgorithm 1) can achieve the optimal convergence rate $O(1/k^{2/3})$, much sharper than the existing one $O(1/k^{1/2})$. Note that prior to our work, to achieve this optimal convergence rate, one needs to develop a very complicated trust region variant with tedious and complex proofs (see Curtis et al, 2017). This is already discussed below Lemma 2.1 of the manuscript and will be highlighted in revision.
>
> The second novelty is to integrate SPIDER estimator with the trust region method which paves a new way for reducing the Hessian variance (more accurate Hessian estimation) in second-order algorithms. As a result, we improve both the first- and second-order computational complexities of trust region method and achieve state-of-the-art computational cost (see Tables 1 and 2).
>
> 2. comparison with Zhou & Gu (2019)：
> We note that Zhou & Gu (2019) has updated their arXiv submission during the review period. As a result, our submission can only compare with the first version of Zhou & Gu (2019).
>
> In the current version, we have compared the results of Zhou & Gu (2019) in Sec. 4.2 and 6.2. For non-Hessian free algorithms in Sec. 4, when we submitted this work, our result is better than theirs under the same assumption: if $1/\epsilon < n < 1/\epsilon^2$, our STR1 outperforms SRVRC; otherwise they have the same complexities (see Sec. 4.2). Note for the case where $n > 1/\epsilon^2$, by further assuming gradient variance is bounded, both methods can achieve the same result. Recently, they improved their algorithm and achieved the same results as ours. As Zhou et al. need to further assume stochastic gradient to be bounded which is absent in ours, for fairness we do not include their result in Table 1. We will mention this difference and then include their results in Table 1 in the revision. For Hessian-free algorithms in Sec. 6, we have compared ours with Zhou & Gu (2019). If $n < 1/\epsilon^2$, then our result is better than theirs; otherwise, we have the same results (see Table 2 and Sec. E.4) with an extra bounded-gradient-variance assumption. In summary, our results for both non-Hessian-free and Hessian-free settings are as good as or stronger than theirs.
>
> Finally, since this paper focuses on stochastic trust region algorithm while Zhou et al. pay attention to stochastic cubic regularization method, we believe both papers are of their own interest and should not be judged only by their complexities.
>
>
> 3. verbose:
> In our revision, we will simplify our presentation and correct the typos.

---

> > ### Comment · AnonReviewer2 · 2019-11-13
> > **Re: Response to reviewer's comments.**
> >
> > Thank you for the response and the clarification. I apologize for the previous comment on the comparison with another paper because I did not notice that their paper has two versions and the version I was referring to was updated after the iclr submission deadline. You can totally ignore my comment on that.

---

> > > ### Author Response · Authors · 2019-11-14
> > > **Re: Re: Response to reviewer's comments.**
> > >
> > > We thank the reviewer for the immediate response. Is there anything else that needs to be clarified?

---

### Decision · Program_Chairs · 2019-12-19

**Decision:**

Reject

**Comment:**

This paper proposes a stochastic trust region method for local minimum finding based on variance reduction, which achieves better oracle complexities than some of the previous work. This is a borderline paper and has been carefully discussed. The main concern of the reviewers is that this paper falls short of proper experiment evaluation to support their theoretical analysis. In detail, the authors proved a globally sublinear convergence rate to a local minimum, yet the experiments demonstrate a linear or even quadratic convergence starting from the initialization. There is a big gap between the theoretical analysis and experiments, which is probably due to the experimental design. In addition, the authors did not submit a revision during the author response (while it is optional), so it is unclear whether a major revision is required to address all the reviewers’ comments. In fact, one reviewer thinks that a major revision is needed. I agree with the reviewers’ evaluation and encourage the authors to improve this paper before future submission.